# The epididymis contributes to sperm DNA integrity and early embryo development through Cysteine-Rich Secretory Proteins

Valeria Sulzyk, Ludmila Curci, Lucas N González, Abril Rebagliati Cid, Mariana Weigel Muñoz*, Patricia S Cuasnicu*

Instituto de Biología y Medicina Experimental (IByME-CONICET), Buenos Aires, Argentina

## eLife Assessment

This **valuable** study reports that epididymal proteins are required for embryogenesis after fertilization. The data presented are generally supportive of the conclusion and considered **solid**. This work will be of interest to reproductive biologists and andrologists.

**\*For correspondence:**
marianaweigel@gmail.com (MWM);
pcuasnicu@gmail.com (PSC)

**Competing interest:** The authors declare that no competing interests exist.

**Abstract** Numerous reports showed that the epididymis plays key roles in the acquisition of sperm fertilizing ability but its contribution to embryo development remains less understood. Female mice mated with males with simultaneous mutations in *Crisp1* and *Crisp3* genes exhibited normal *in vivo* fertilization but impaired embryo development. In this work, we found that this phenotype was not due to delayed fertilization, and it was observed in eggs fertilized by epididymal sperm either *in vivo* or *in vitro*. Of note, eggs fertilized *in vitro* by mutant sperm displayed impaired meiotic resumption unrelated to $Ca^{2+}$ oscillations defects during egg activation, supporting potential sperm DNA defects. Interestingly, cauda but not caput epididymal mutant sperm exhibited increased DNA fragmentation, revealing that DNA integrity defects appear during epididymal transit. Moreover, exposing control sperm to mutant epididymal fluid or to $Ca^{2+}$-supplemented control fluid significantly increased DNA fragmentation. This, together with the higher intracellular $Ca^{2+}$ levels detected in mutant sperm, supports a dysregulation in $Ca^{2+}$ homeostasis within the epididymis and sperm as the main factor responsible for embryo development failure. These findings highlight the contribution of the epididymis beyond fertilization and identify CRISP1 and CRISP3 as novel factors essential for sperm DNA integrity and early embryo development.

## Introduction

Mammalian sperm that leave the testes do not have the ability to fertilize an egg and must undergo different processes to become fertilizing competent. Initially, they need to mature while they pass through the epididymis which provides a suitable environment for acquisition of fertilizing ability, storage, and protection of sperm (*Robaire and Hinton, 2015*). Subsequently, sperm must undergo another process known as capacitation while ascending through the female reproductive tract which will allow them to undergo the acrosome reaction in the head and to develop a hyperactivated motility in the tail, both essential for fertilization (*Florman and Fissore, 2015*). Once they reach the site of fertilization in the oviduct, sperm penetrate the different coats that surround the egg (i.e. cumulus oophorus and zona pellucida (ZP)) and fuse with the plasma membrane of the egg, resulting in the

formation of a single-cell zygote which then embarks on its development into a multicellular organism through a highly complex and tightly regulated process (*Ramathal et al., 2015*). Whereas numerous reports show the relevance of post-testicular maturation for fertilization, less information exists on the contribution of epididymal transit to embryo development.

The epididymis, the main location where sperm maturation takes place, has a very specialized epithelium involved in ion transport, protein matrix formation, and both protein and vesicle secretion (*Robaire and Hinton, 2015*). Among such secretory proteins are CRISP (Cysteine-RIch Secretory Proteins), a group of evolutionarily conserved proteins mainly expressed in the male reproductive tract (*Gibbs et al., 2008*; *Gonzalez et al., 2021*). In mammals, four members have been described which escort sperm during their transit through both the male and female reproductive tracts. CRISP are characterized by the presence of 16 conserved Cys and two domains that have evolved to perform a variety of functions. Whereas the N-terminal domain of CRISP proteins (i.e. PR-1) is involved in cell-to-cell interaction (*Ellerman et al., 2006*; *Maeda et al., 1998*), proteolytic processes (*Milne et al., 2003*) and amyloid-like-aggregation activity (*Olrichs et al., 2014*; *Sheng et al., 2019*), the C-terminal domain (i.e. CRD) has the ability to regulate ion channels, such as ryanodine (RyR), CNGs, TRPM8, and Catsper (*Ernesto et al., 2015*; *Gibbs et al., 2006*; *Gibbs et al., 2011*; *Yamazaki et al., 2002*).

Biochemical, molecular, and genetic approaches from our group and others revealed that CRISP proteins play key roles in sperm maturation, capacitation and fertilization (*Gonzalez et al., 2021*). Epididymal CRISP1, first described by our laboratory (*Cameo and Blaquier, 1976*, *Cohen et al., 2000a*) associates with sperm during maturation and both inhibits CatSper (*Ernesto et al., 2015*), the primary sperm $Ca^{2+}$ channel essential for male fertility (*Ren et al., 2001*; *Sun et al., 2017*), and participates in gamete interaction through egg-complementary sites (*Busso et al., 2007*; *Cohen et al., 2000b*; *Rochwerger et al., 1992*). CRISP4, almost exclusively synthesized in the epididymis, also associates with sperm during maturation (*Jalkanen et al., 2005*; *Nolan et al., 2006*; *Weigel Muñoz et al., 2019*), inhibits $Ca^{2+}$ channel TRPM8 (*Gibbs et al., 2011*), and participates in the fertilization process (*Carvajal et al., 2018*; *Gibbs et al., 2011*; *Turunen et al., 2012*). CRISP3 exhibits a wider expression distribution including the epididymis and accessory glands within the male reproductive tract (*Haendler et al., 1997*; *Schwidetzky et al., 1995*; *Udby et al., 2005*) as well as organs and cells with immunological functions (*Reddy et al., 2008*). Whereas CRISP3 was found to bind to sperm with no roles in fertilization described so far (*Da Ros et al., 2015*), several reports support its association with male fertility in different species including humans (*Chen et al., 2020*; *Doty et al., 2011*; *Gholami et al., 2021*).

In spite of clear evidence supporting the relevance of CRISP proteins for fertilization, knockout (KO) male mice for each individual CRISP remain fertile (*Brukman et al., 2016*; *Carvajal et al., 2018*; *Da Ros et al., 2008*; *Gibbs et al., 2011*; *Turunen et al., 2012*; *Volpert et al., 2020*; *Weigel Muñoz et al., 2018*), suggesting the existence of compensatory mechanisms among homologous CRISP family members. This idea was later confirmed by experiments from our group showing that simultaneous modifications in more than one CRISP gene can significantly impair male fertility (*Carvajal et al., 2018*). Whereas males lacking epididymal CRISP1 and CRISP4 (C1/C4 DKO) were subfertile due to a significant decrease in *in vivo* fertilization (*Carvajal et al., 2018*), males lacking CRISP1 and CRISP3 (C1/C3 DKO) were subfertile but exhibit normal *in vivo* fertilization (*Curci et al., 2020*), supporting the notion that fertility inhibition in this colony resulted as a consequence of post-fertilization defects. Interestingly, subfertility in C1/C3 DKO colony was associated with a failure of *in vivo* fertilized eggs to reach the blastocyst stage, revealing the potential relevance of CRISP1 and CRISP3 for early embryo development (*Curci et al., 2020*). Moreover, examination of ejaculated sperm within the uterus of control mated females showed that while control sperm were freely moving in the uterine fluid, C1/C3 DKO sperm were mostly immotile and trapped into aggregates within a viscous uterine fluid (*Curci et al., 2020*), suggesting the relevance of these two proteins for sperm survival within the female reproductive tract. Considering the expression of CRISP1 and CRISP3 in the male reproductive tract, it is possible that defects during and/or following epididymal passage are responsible for the early embryo development defects observed for the mutant males. In view of this, in the present work, we explored possible mechanisms leading to C1/C3 DKO male phenotype and provide novel evidence supporting the contribution of the epididymis and the relevance of both CRISP1 and CRISP3 for sperm DNA integrity and early embryo development.

**Table 1.** Analysis of embryo development corresponding to C1/C3 and C1/C4 DKO males.

|  | Control | C1/C3 DKO | Control | C1/C4 DKO |
| --- | --- | --- | --- | --- |
| Total two-cell embryos (N°) | 53 | 53 | 80 | 76 |
| Blastocysts (N°) | 38 | 23 | 52 | 46 |
| Embryo development (%) | 75.2 ± 8.3% | 37.4 ± 10.7%** | 67.7 ± 12.1% | 66.9 ± 13.4% |

Note: Percentage of embryo development was calculated as the mean of at least five independent experiments, *n* = 5. **p < 0.01 vs control.

The online version of this article includes the following source data for table 1:

**Source data 1.** Raw data and statistical analisys for C1/C3 DKO *in vivo* fertilization.

**Source data 2.** Raw data and statistical analisys for C1/C3 DKO embryo development.

**Source data 3.** Raw data and statistical analisys for C1/C4 DKO *in vivo* fertilization.

**Source data 4.** Raw data and statistical analisys for C1/C4 DKO embryo development.

## Results

### Embryo development defects associated with the lack of CRISP1 and CRISP3 are not due to a delayed fertilization

To analyze whether defects in early embryo development were specifically linked to mutations in C*risp1* and *Crisp3* genes or could also be contributing to the subfertility of C1/C4 DKO males (*Carvajal et al., 2018*), superovulated females were mated with mutant or control males from each DKO colony and those eggs recovered from the ampulla and reaching the two-cell stage *in vitro* (i.e. fertilized eggs) were further incubated to analyze the percentage progressing to blastocysts. Results revealed that whereas two-cell embryos from the C1/C3 DKO group showed a significant decrease in the percentage of blastocysts as previously reported (*Curci et al., 2020*), those corresponding to C1/C4 DKO mice showed no differences in the percentage of blastocyst compared to controls (*Table 1*), supporting that early embryo development defects were caused by the specific simultaneous mutation of *Crisp1* and *Crisp3* genes.

Considering that delays in the time of *in vivo* fertilization could lead to embryonic development defects (*Brackett et al., 1978*; *Lacham and Trounson, 1991*; *Lacham-Kaplan and Trounson, 1994*; *Orgebin-Crist, 1968*; *Orgebin-Crist and Jahad, 1977*), and given the presence of aggregates of immotile sperm in the uterus of females mated with C1/C3 DKO males (*Curci et al., 2020*), we next investigated whether the early embryo development failure in this colony was due to a delayed fertilization caused by an impaired sperm transport within the female reproductive tract. To this aim, we analyzed both sperm migration within the oviduct and *in vivo* fertilization shortly after mating (i.e. 4 hr) as a way to avoid the possibility that the prolonged stay of sperm within the female tract corresponding to the conventional mating schedule (18 hr), could be giving defective sperm enough time to reach the ampulla and fertilize the eggs. Using Acrosine-GFP (Green Fluorescent Protein)-tagged C1/C3 DKO or control males, we first examined sperm migration within the oviduct via fluorescence microscopy 4 hr after observation of copulatory plugs. Results indicated that both mutant and control sperm exhibited no difficulties to pass the uterotubal junction and migrate within the oviduct as judged by the presence of labeled sperm in both the lower and middle isthmus (*Figure 1*). The observation of very few fluorescent mutant or control sperm beyond the isthmus (*Figure 1B, C*) is due to the reported loss of the acrosome in sperm after reaching the middle/upper isthmus (*La Spina et al., 2016*; *Muro et al., 2016*). Consistent with these observations, examination of fertilization in the ampulla 4 hr after mating showed no significant differences between groups in the percentage of eggs recovered from the ampulla that develop to two-cell embryos *in vitro* (*Figure 2A*), confirming no defects in the time of arrival of sperm to the ampulla. However, once again, the eggs fertilized by mutant sperm exhibited clear defects to reach the blastocyst stage *in vitro* compared to controls (*Figure 2B*). Together, our observations suggest that factors other than a delayed fertilization due to transport defects were responsible for the observed embryo development phenotype of the mutant colony.

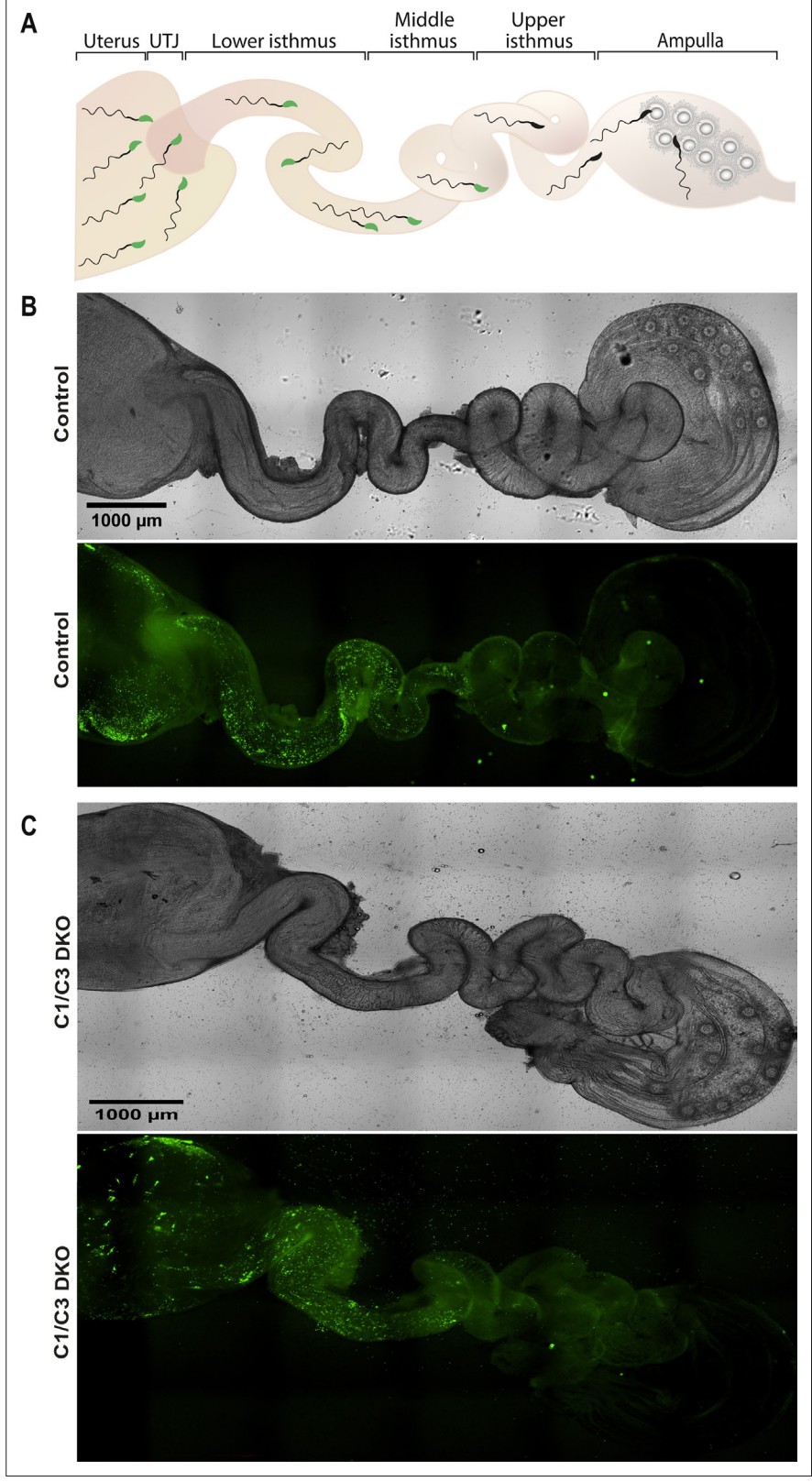

**Figure 1.** Sperm migration through the female reproductive tract. (**A**) Representative diagram showing sperm within the different regions of the female reproductive tract. UTJ, utero-tubal junction. Superovulated females were mated with Acrosine-GFP control or C1/C3 DKO males and, 4 hr after mating, sperm were analyzed inside the tract by fluorescence microscopy. (**B**) Bright field (upper panel) and fluorescence (lower panel) images of the

*Figure 1 continued on next page*

*Figure 1 continued*

reproductive tract of a female mated with a control male (×40). (**C**) Bright field (upper panel) and fluorescence (lower panel) images of the reproductive tract of a female mated with C1/C3 DKO male (×40). Figures are representative of at least three independent experiments.

The online version of this article includes the following source data for figure 1:

**Source data 1.** Raw DIC picture of the reproductive tract of a female mouse mated with a control male (*Figure 1B*).

**Source data 2.** Raw fluorescence picture of the reproductive tract of a female mouse mated with a control male (*Figure 1B*).

**Source data 3.** Raw DIC picture of the reproductive tract of a female mouse mated with a C1C3 DKO male (*Figure 1C*).

**Source data 4.** Raw fluorescence picture of the reproductive tract of a female mouse mated with a C1C3 DKO male (*Figure 1C*).

## Mutant epididymal sperm already carry defects leading to embryo development failure

Given that our results had been obtained by natural mating, we next investigated whether the embryo development defects observed for C1/C3 DKO males appear during or after epididymal transit. To address this question, we inseminated C1/C3 DKO mature cauda epididymal sperm into one uterine horn and control cauda sperm in the contralateral horn of superovulated females, and then analyzed the percentage of eggs recovered from the ampulla capable of reaching the two-cell and blastocyst stages *in vitro*. Results showed that although no differences between groups were observed in the percentage of two-cell embryos (*Figure 2C*), the percentage of two-cell embryos progressing to blastocysts was significantly lower for mutant than for control sperm (*Figure 2D*), revealing that sperm defects contributing to embryo development deficiencies in C1/C3 DKO males were already present at the epididymal level.

It is known that fertilization under *in vitro* conditions provides a controlled environment for gamete interaction, avoiding potential sperm selection mechanisms or delays in sperm arrival to the egg that may occur *in vivo*, and allowing a more precise analysis of the kinetics of both fertilization and embryo development. In view of this, we next conducted a series of *in vitro* fertilization (IVF) studies using capacitated epididymal sperm and eggs surrounded or devoid of their coats. Results showed that whereas no significant differences in either fresh or capacitated sperm were found in sperm count, viability, and progressive motility between groups (*Supplementary file 1*), co-incubation of cumulus–oocyte complexes (COC) with mutant sperm led to significantly lower percentages of both fertilized eggs evaluated by Hoechst staining (i.e. decondensing heads or two pronuclei within the ooplasm) (*Figure 3A*) and eggs capable of progressing to two-cell embryos (*Figure 3B*). Interestingly, besides these fertilization defects, when two-cell embryos continued their incubation *in vitro* and the percentage of embryos at different stages of development was analyzed, a significant decrease in the percentage of embryos reaching the morula and blastocyst stages was observed for the mutant group (*Figure 3C*), confirming that defects leading to embryo development failure were already present in epididymal cells and could be detected even outside the female tract environment.

To investigate whether difficulties in penetration of the egg coats that surround the egg could generate a potential delay in fertilization that finally leads to embryo development failure, IVF assays were carried out using eggs devoid of both cumulus cells and ZP, and the percentage of fertilized ZP-free eggs analyzed. Under these conditions, there was a lower but still significant decrease in the percentage of eggs fertilized by mutant sperm accompanied again by significantly lower rates of blastocysts (*Figure 3D, E*), indicating that defects in egg coat penetration were not responsible for embryo development failure.

To further analyze the mechanisms leading to embryo development defects, ZP-free eggs were co-incubated with capacitated sperm as above and both sperm and egg DNA status within the ooplasma of fertilized eggs were analyzed by Hoechst staining. Of note, results showed that whereas all eggs with decondensing heads had already extruded the 2nd polar body in controls, in four out of six experiments, a proportion of eggs with decondensing heads corresponding to the mutant group

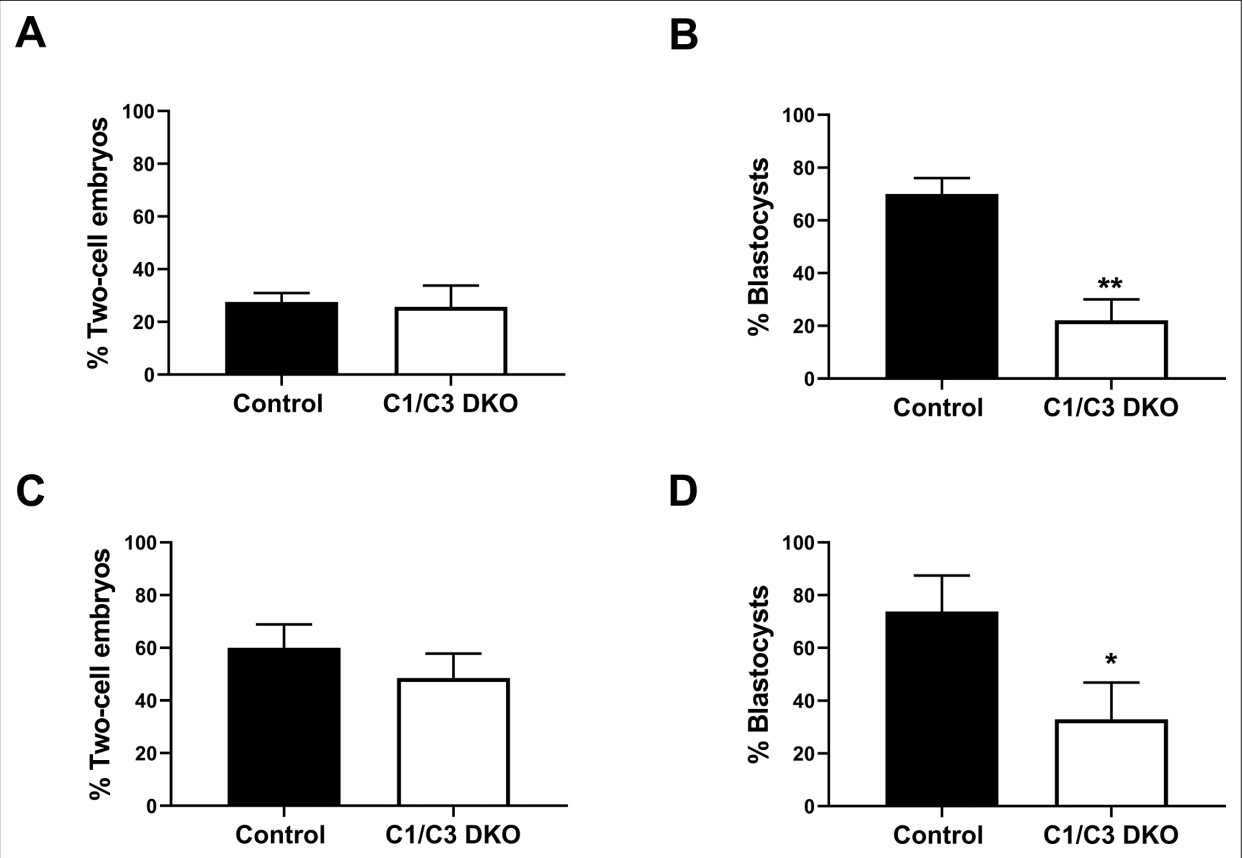

**Figure 2.** *In vivo* fertilization and embryo development. (**A**) Control and C1/C3 DKO males were mated with superovulated females and the percentage of fertilized eggs recovered from the ampulla 4 hr after mating was evaluated. Eggs were considered fertilized when they reached the two-cell embryo stage 24 hr after *in vitro* incubation. (**B**) Two-cell embryos from (**A**) were incubated *in vitro* for an additional 3 days, and the percentage reaching the blastocyst stage was determined. (**C**) Control or C1/C3 DKO cauda epididymal sperm were inseminated into the uterus of superovulated females. After 15 hr, eggs were recovered from the ampulla, incubated for 24 hr *in vitro* and were considered fertilized when reaching the two-cell embryo stage. (**D**) Two-cell embryos from (**C**) were incubated *in vitro* for an additional 3 days and the percentage reaching the blastocyst stage was determined. Data are mean ± SEM; $n = 5$, *$p < 0.05$; **$p < 0.01$. Percentages of two-cell embryos were determined by dividing the number of two-cell embryos by the total number of eggs examined and percentages of blastocysts as the number of eggs reaching the blastocyst stage divided by the total number of two-cell embryos recovered.

The online version of this article includes the following source data for figure 2:

**Source data 1.** Raw data and statistical analysis of the percentage of fertilized eggs recovered from the ampulla 4 hr after copulatory plug formation (*Figure 2A*).

**Source data 2.** Raw data and statistical analysis of the percentage of embryos reaching the blastocyst stage *in vitro* from two-cell embryos obtained after *in vitro* incubation of fertilized eggs recovered from the ampulla 4hs after copulatory plug formation (*Figure 2B*).

**Source data 3.** Raw data and statistical analysis of the percentage of fertilized egg recovered from the ampulla after intrauterine insemination of cauda epididymal sperm (*Figure 2C*).

**Source data 4.** Raw data and statistical analysis of the percentage of embryos reaching the blastocyst stage *in vitro* from two-cell embryos obtained after *in vitro* incubation of eggs recovered from the ampulla after intrauterine insemination of cauda epididymal sperm (*Figure 2D*).

were still at Metaphase II (Met II) (*Figure 3F*), revealing defects in epididymal sperm affecting early post-fertilization events as the potential cause of the phenotype observed in the mutant colony.

## Mutant epididymal sperm exhibited higher levels of both DNA fragmentation and intracellular Ca²⁺

The finding that a proportion of eggs fertilized by epididymal mutant sperm *in vitro* were still at Met II, opened the possibility of defects in the meiotic resumption event that occurs during egg activation. Based on this, we next monitored the characteristic repetitive series of changes in intracellular Ca²⁺

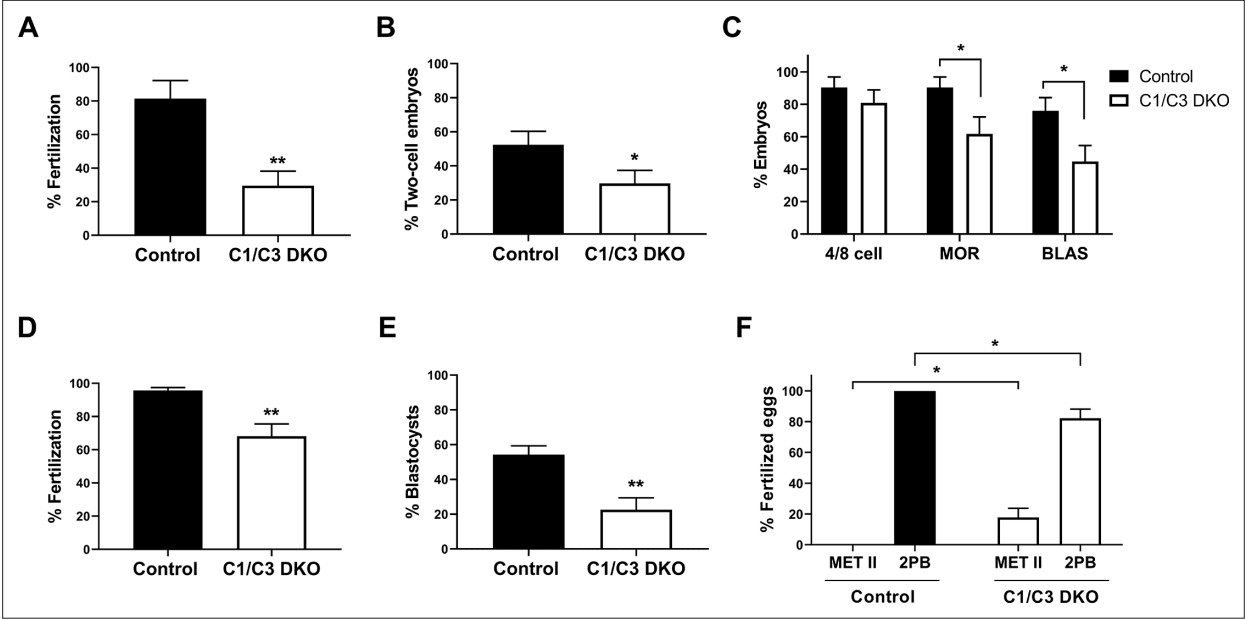

**Figure 3.** *In vitro* fertilization and embryo development. (**A**, **B**) *In vitro* capacitated control or C1/C3 DKO cauda epididymal sperm were co-incubated with cumulus–oocyte complexes (COC) for 3.5 hr. Eggs were either analyzed at that moment and considered fertilized when at least one decondensing sperm nucleus or two pronuclei were found in the ooplasm (*n* = 3) (**A**) or incubated for additional 24 hr to determine the percentage reaching the two-cell embryo stage (*n* = 7) (**B**). (**C**) Two-cell embryos from (**B**) were incubated for 3 days *in vitro* and the percentage reaching 4/8 cells (day 1), morula (day 2), or blastocyst (day 3) stages determined (*n* = 7). (**D**) *In vitro* capacitated control or C1/C3 DKO epididymal sperm were co-incubated with zona pellucida (ZP)-free eggs for 1 hr and fertilization was evaluated by DNA staining. Eggs were considered fertilized when at least one decondensing sperm nucleus was found in the ooplasm (*n* ≥ 5). (**E**) Fertilized ZP-free eggs obtained as in D were incubated for an additional 3 days *in vitro* and the percentage progressing to blastocysts was determined (*n* = 5). (**F**) Fertilized eggs from (**D**) were analyzed for maternal DNA status and classified as arrested in Metaphase II (Met II) or exhibiting 2nd polar body (2PB) (*n* ≥ 5). Data are the mean ± SEM; *p < 0.05; **p < 0.01. Percentages of fertilization were determined by dividing the number of fertilized eggs by the total number of eggs examined. Percentages of two-cell embryos were determined by dividing the number of two-cell embryos by the total number of eggs examined. Percentages of either 4/8 cell embryos, morula and blastocysts were calculated as the number of embryos reaching each of these stages divided by the total number of two-cell embryos obtained.

The online version of this article includes the following source data for figure 3:

**Source data 1.** Raw data and statistical analysis of the percentage of eggs exhibiting a decondensed sperm head after *in vitro* co-incubation of cumulus–oocyte complexes (COC) with cauda epididymal sperm (*Figure 3A*).

**Source data 2.** Raw data and statistical analysis of the percentage of two-cell embryos after *in vitro* co-incubation of cumulus–oocyte complexes (COC) and cauda epididymal sperm (*Figure 3B*).

**Source data 3.** Raw data and statistical analysis of the percentage of embryos reaching the blastocyst stage *in vitro* from two-cell eggs obtained after *in vitro* co incubation of cumulus–oocyte complexes (COC) and cauda epididymal sperm (*Figure 3C*).

**Source data 4.** Raw data and statistical analysis of the percentage of eggs exhibiting a decondensed sperm head after *in vitro* co-incubation of zona pellucida-free oocytes with cauda epididymal sperm (*Figure 3D*).

**Source data 5.** Raw data and statistical analysis of the percentage of embryos reaching the blastocyst stage *in vitro* from two-cell eggs obtained after *in vitro* co-incubation of zona pellucida-free oocytes with cauda epididymal sperm (*Figure 3E*).

**Source data 6.** Raw data and statistical analysis of maternal DNA status after *in vitro* co-incubation of zona pellucida-free oocytes with cauda epididymal sperm (*Figure 3F*).

concentration ($Ca^{2+}$ oscillations) known to underpin release from meiotic arrest during egg activation and initiation of embryo development in mammalian eggs (*Miyazaki, 2006*; *Wakai et al., 2019*). For this purpose, ZP-free eggs were stained with Fluo-4 AM, co-incubated *in vitro* with control or mutant capacitated epididymal sperm and subjected to live confocal fluorescence imaging. Results showed no differences between the two genotypes in either the pattern of $Ca^{2+}$ oscillations (*Figure 4A*) or in a series of associated parameters such as the number of oscillations within 90 min, time until first oscillation (considered the time to gamete fusion), oscillation frequency, first transient area or first transient duration (*Figure 4B–F*). Together, these observations indicate that the presence of eggs still at Met II

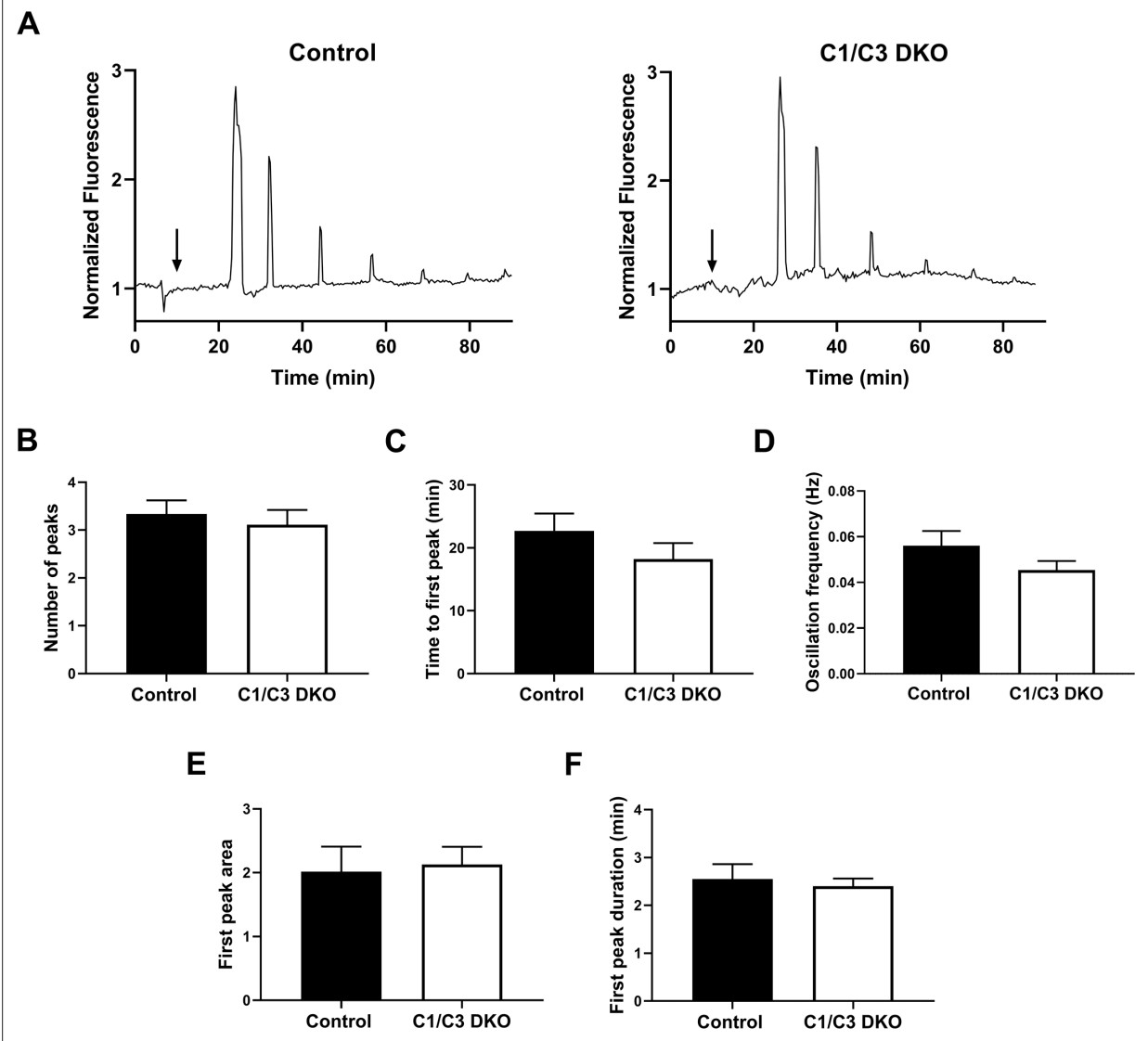

**Figure 4.** Egg $Ca^{2+}$ oscillations in *in vitro* fertilized eggs. (**A**) Representative traces of $Ca^{2+}$ oscillation patterns following *in vitro* fertilization of zona pellucida (ZP)-free eggs with control or C1/C3 DKO sperm. Arrows indicate the time of sperm addition. (**B**) Number of peaks within 90 min, (**C**) time to first peak, (**D**) oscillation frequency, (**E**) first peak area under the curve, and (**F**) first transient duration. Values were normalized to basal $Ca^{2+}$ levels recorded prior to sperm addition. Data are mean ± SEM of at least nine oocytes from three independent experiments; (**A–F**) ns.

The online version of this article includes the following source data for figure 4:

**Source data 1.** Raw data and statistical analysis of the numbers of peaks corresponding to $Ca^{2+}$ oscillations analyzed by fluorescence microscopy (*Figure 4B*).

**Source data 2.** Raw data and statistical analysis of the time to the first peak corresponding to $Ca^{2+}$ oscillations analyzed by fluorescence microscopy (*Figure 4C*).

**Source data 3.** Raw data and statistical analysis of the frequency of $Ca^{2+}$ oscillations analyzed by fluorescence microscopy (*Figure 4D*).

**Source data 4.** Raw data and statistical analysis of the first peak area corresponding to $Ca^{2+}$ oscillations analyzed by fluorescence microscopy (*Figure 4E*).

**Source data 5.** Raw data and statistical analysis of the duration of the first peak corresponding to $Ca^{2+}$ oscillations analyzed by fluorescence microscopy (*Figure 4F*).

among those fertilized by mutant sperm was not due to defects in $Ca^{2+}$ dynamics known to be critical for meiotic resumption during egg activation.

Considering that delays in early embryo development may result from the time taken by the egg to repair damaged paternal DNA (*Esbert et al., 2018*; *Newman et al., 2022*; *Nguyen et al., 2023*), we

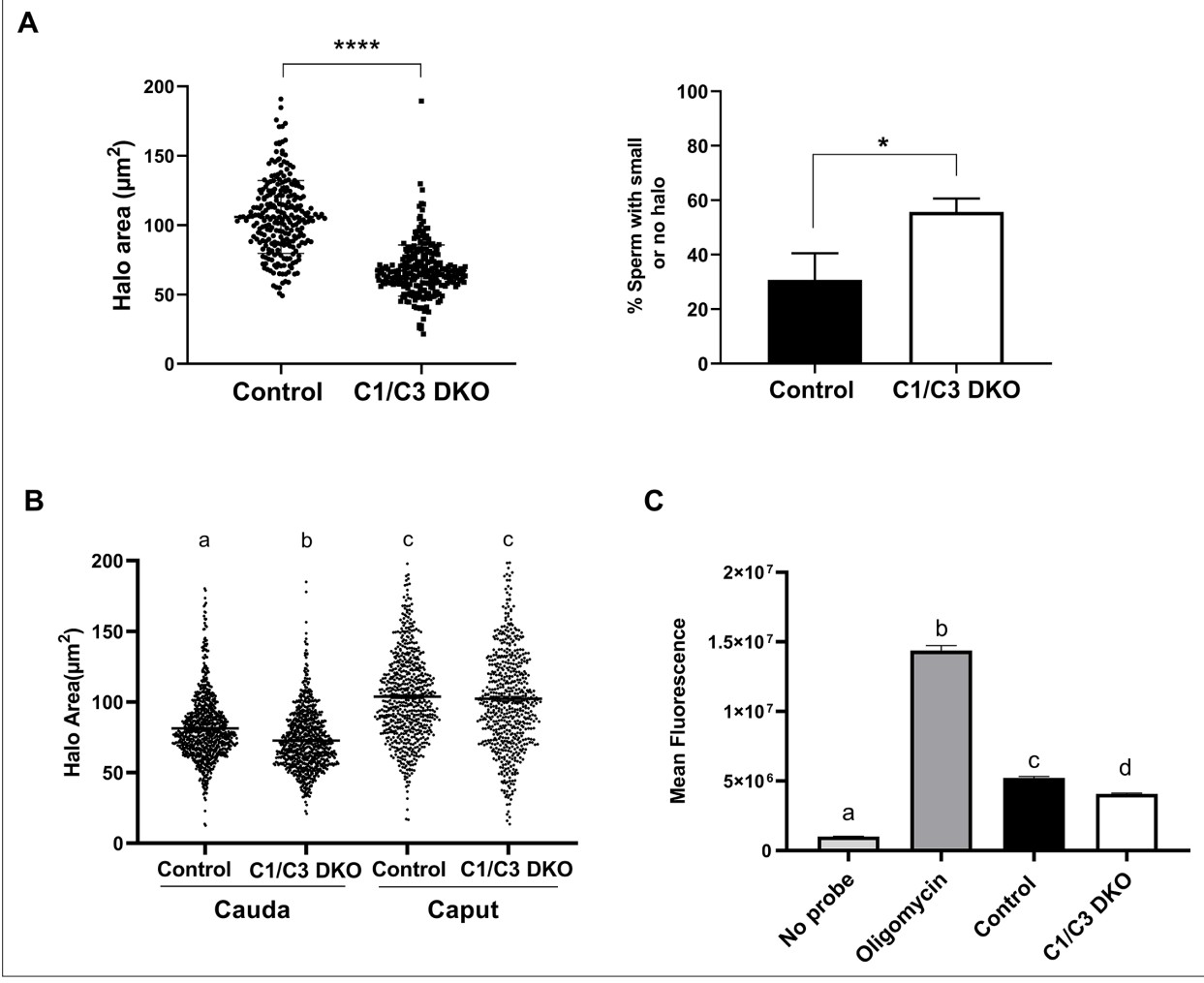

**Figure 5.** Sperm DNA fragmentation and reactive oxygen species (ROS) levels. (**A**) DNA fragmentation of control or C1/C3 DKO cauda epididymal sperm was analyzed by sperm chromatin dispersion (SCD) assay and both the area of DNA halo in each individual cell (left panel) and the percentage of sperm heads with small or no halo (right panel) determined. (**B**) DNA fragmentation of cauda and caput epididymal control or C1/C3 DKO sperm was analyzed by SCD assay and the area of DNA halo in each individual cell determined. (**C**) ROS levels in control and C1/C3 DKO cauda sperm analyzed by fluorescence confocal microscopy. Absence of probe and presence of Oligomycin were used as negative and positive controls, respectively. In all cases, $n = 4$. *$p < 0.05$; ****$p < 0.0001$, different letters indicate significant differences between treatments, $p < 0.0001$.

The online version of this article includes the following source data for figure 5:

**Source data 1.** Raw data and statistical analysis of the DNA halo area corresponding to sperm analyzed by sperm chromatin dispersion (SCD) assay (*Figure 5A*).

**Source data 2.** Raw data and statistical analysis of the percentage of DNA fragmentation corresponding to cauda sperm analyzed by sperm chromatin dispersion (SCD) assay (*Figure 5A*).

**Source data 3.** Raw data and statistical analysis of the DNA halo area corresponding to caput and cauda sperm analyzed by SCD assay (*Figure 5B*).

**Source data 4.** Raw data and statistical analysis of reactive oxygen species (ROS) levels in fresh cauda sperm analyzed by fluorescence microscopy (*Figure 5C*).

next decided to analyze possible defects in DNA integrity in mutant cauda epididymal sperm. These studies were carried out using the sperm chromatin dispersion (SCD) assay which is based on the principle that sperm with fragmented DNA fail to produce the characteristic halo of dispersed DNA loops observed in sperm with non-fragmented DNA (*Fernández et al., 2003*). Interestingly, results showed a significantly higher level of DNA fragmentation in mutant than control sperm as indicated by the distribution of individual cells as a function of their DNA halo area and the percentage of cells failing to produce the halo (*Figure 5A*). To investigate whether the higher DNA fragmentation levels observed in mutant cauda sperm appear during epididymal transit, DNA fragmentation was

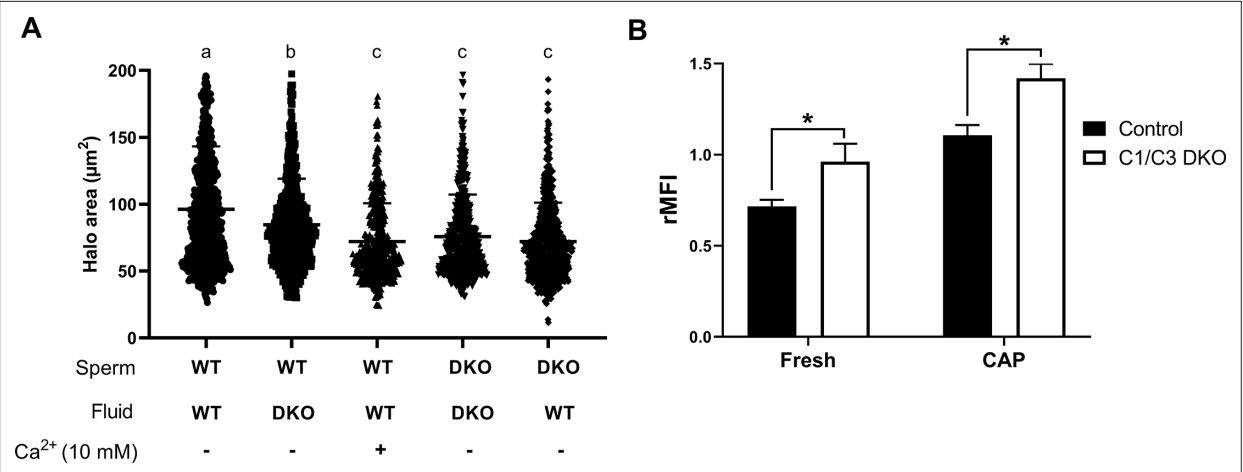

**Figure 6.** Influence of epididymal fluid on sperm DNA integrity and intracellular $Ca^{2+}$ levels. (**A**) DNA fragmentation of control or C1/C3 DKO cauda epididymal sperm incubated for 1 hr with their own or the other genotype epididymal fluid in the presence or absence of 10 mM $Ca^{2+}$ was analyzed by sperm chromatin dispersion (SCD) assay and the area of DNA halo in each individual cell determined. Different letters indicate significant differences between treatments (p < 0.05). (**B**) Intracellular $Ca^{2+}$ levels were evaluated by flow cytometry using Fluo-4-AM probe. Results are shown as normalized mean fluorescence intensity (rMFI) of Fluo-4-AM compared to the control condition in each experiment for non-capacitated (fresh) and capacitated (CAP) sperm. Data are mean ± SEM, *n* = 5; *p < 0.05.

The online version of this article includes the following source data for figure 6:

**Source data 1.** Raw data and statistical analysis of the DNA halo area corresponding to sperm incubated *in vitro* with epididymal fluids and $Ca^{2+}$ (*Figure 6A*).

**Source data 2.** Raw data and statistical analysis of the normalized intensity of Fluo-4-AM fluorescence obtained by flow cytometry of fresh and capacitated cauda epididymal sperm (*Figure 6B*).

also analyzed in immature sperm recovered from the caput region using cauda sperm from the same epididymis as control. Results showed that caput cells exhibited no differences in DNA fragmentation levels between groups, indicating that defects in DNA integrity in mutant sperm develop within the epididymis (*Figure 5B*). Given the well-established relationship between DNA fragmentation and oxidative stress, we next analyzed reactive oxygen species (ROS) levels in cauda epididymal sperm from control and mutant mice. Interestingly, results showed that ROS levels in mutant sperm were not higher than those observed in the control group (*Figure 5C*), not favoring the idea that higher sperm DNA fragmentation levels were due to an increase in oxidative stress in mutant cells.

To assess the impact of the epididymal environment on sperm DNA fragmentation, WT cauda epididymal sperm were exposed to epididymal fluid recovered from mutant mice, and sperm DNA integrity analyzed by the SCD assay. Results showed that, under these conditions, WT sperm exhibited significantly higher levels of DNA fragmentation, supporting the relevance of epididymal fluid for sperm DNA integrity. Exposure of mutant sperm to epididymal fluid from WT mice, on the other hand, did not modify the higher DNA fragmentation levels exhibited by mutant cells (*Figure 6A*). Considering both reports showing that sperm DNA fragmentation can be induced by divalent cations in the presence of epididymal fluids (*Gawecka et al., 2015*; *Shaman et al., 2006*), and that CRISP proteins are $Ca^{2+}$ channel regulators (*Ernesto et al., 2015*; *Gibbs et al., 2006*; *Gibbs et al., 2011*), we next analyzed the possibility that higher $Ca^{2+}$ levels in the epididymis might contribute to the impaired DNA integrity of mutant sperm. For this purpose, WT cauda epididymal sperm were incubated *in vitro* with WT epididymal fluid in the presence of $Ca^{2+}$ (10 mM), detecting significantly higher levels of DNA fragmentation in these cells compared to controls incubated in the absence of the cation (*Figure 6A*). As another approach to explore the involvement of $Ca^{2+}$ in sperm DNA fragmentation, we next analyzed intracellular $Ca^{2+}$ in cauda epididymal sperm by flow cytometry detecting significantly higher $Ca^{2+}$ levels in mutant than control sperm either before or after capacitation (*Figure 6B*). Together, these observations support a dysregulation of $Ca^{2+}$ within the epididymis and sperm as the main responsible for the higher sperm DNA fragmentation levels and, thus, the subsequent embryo development failure observed for mutant males.

## Discussion

Substantial evidence supports the involvement of the epididymis in the acquisition of sperm fertilizing ability (*Björkgren and Sipilä, 2019*). However, limited information exists on the contribution of epididymal transit to the early stages of embryo development known to be crucial in determining the overall fitness of the organism. In this regard, whereas the molecular mechanisms underlying the impact of paternal factors on embryo development remain to be fully elucidated, the present work provides novel findings supporting the contribution of the epididymis to early embryo development and identifying CRISP1 and CRISP3 proteins as novel male factors relevant to this process.

Previous observations from our laboratory (*Curci et al., 2020*) showed that females mated with males with mutations in *Crisp1* and *Crisp3* genes exhibited normal *in vivo* fertilization rates but significantly lower percentages of embryo development, supporting the notion that male-derived factors are required to allow for correct development of the embryo (*Vallet-Buisan et al., 2023*). Moreover, we observed that triple knockouts lacking CRISP1, CRISP2, and CRISP3 as well quadruple knockouts lacking the four members of the family exhibited both fertilization and embryo development defects, supporting the lack of CRISP1 and CRISP3 as relevant for the embryo development phenotype. Expanding on those observations, in the present study we confirmed an association between embryo development failure and mutations in *Crisp1* and *Crisp3* as indicated by the finding that eggs fertilized by sperm from males lacking *Crisp1* and *Crisp4* genes did not exhibit evidence of embryo development failure. Thus, whereas males from both double knockout colonies are subfertile and share the lack of CRISP1, the mechanisms underlying their subfertility are different depending on the other simultaneously deleted protein, providing information on the different functional modules that operate within the CRISP family (*Curci et al., 2020*; *Gonzalez et al., 2021*), and establishing two models that uncouple fertilization from early embryo development. In this way, whereas C1/C4 DKO mice exclusively exhibiting defects in fertilization contribute to a better understanding of the mechanisms involved in gamete interaction, C1/C3 DKO males showing only defects in egg progression to blastocysts become an excellent model for elucidating the molecular mechanisms inherent to the early embryo development process.

Our findings showing the presence of aggregates of immotile sperm within the uterus of females mated with C1/C3 DKO males (*Curci et al., 2020*) opened the possibility that a delayed fertilization due to a retarded arrival of sperm to the ampulla (the site of fertilization) could be the reason for embryo development failure as previously reported for immature sperm (*Brackett et al., 1978*; *Lacham and Trounson, 1991*; *Lacham-Kaplan and Trounson, 1994*; *Orgebin-Crist, 1968*; *Orgebin-Crist and Jahad, 1977*). Evaluation of sperm migration within the female tract shortly after mating to avoid a delayed fertilization showed, however, that mutant sperm could pass through the uterotubal junction and reach the lower/middle isthmus just like control sperm, indicating that embryo development failure in mutant mice is unlikely to be due to defects in sperm transport from the uterus to the oviduct and/or migration within the oviduct. Moreover, although the loss of the acrosomes due the occurrence of the acrosome reaction in the isthmus (*La Spina et al., 2016*; *Muro et al., 2016*) did not allow the detection of labeled sperm within the ampulla, our results indicated that the time of sperm arrival to the ampulla was comparable to that of control sperm as evidenced by fertilization rates not different from controls at 4 hr post copulation. However, in spite of the lack of differences in sperm arrival to the ampulla and fertilization rates observed shortly after mating, the fertilized eggs still exhibited an impaired ability to reach the blastocyst stage, not favoring the idea that a delayed fertilization and/or a consequently compromised oocyte quality could be responsible for the early embryo development defects observed in the mutant colony.

Considering various reports showing that male accessory glands contribute to proper embryo development without affecting fertilization rates (*Jodar, 2019*; *Ma et al., 2022*; *Vallet-Buisan et al., 2023*; *O et al., 1988*), the possibility existed that the sperm defects leading to the embryo development phenotype were associated with the lack of CRISP1 and CRISP3 in the accessory gland secretions. Interestingly, however, when epididymal sperm were directly inseminated into the uterus, we observed again normal fertilization rates in the ampulla accompanied by impaired embryo development, revealing that defects were already present in epididymal sperm. This conclusion was further supported by IVF studies showing that cumulus oocyte complexes fertilized by mutant epididymal sperm also exhibited embryo development defects. Unlike *in vivo* assays, however, IVF studies revealed fertilization defects in mutant sperm not detected by mating. This difference is likely due to

the effective mechanism of sperm selection within the female reproductive tract that allows only high-quality sperm to reach the eggs (*Cummins and Yanagimachi, 1982*) even in males with reproductive deficiencies, thus, masking sperm defects detected under *in vitro* conditions (*Brukman et al., 2016*). Nevertheless, the finding that C1/C3 DKO sperm exhibit IVF capabilities similar to those reported for sperm lacking only CRISP1 (*Da Ros et al., 2008*), does not support a major role for murine CRISP3 in gamete interaction in agreement with our previous observations for human CRISP3 (*Da Ros et al., 2015*).

The observation that ZP-free oocytes fertilized *in vitro* by mutant epididymal sperm also exhibited a lower ability to reach the blastocyst stage excluded potential delays generated by deficiencies in egg coat penetration as a cause of embryo development failure, indicating the existence of defects other than those associated with gamete interaction *per se* as responsible for the mutant colony phenotype. Interestingly, a proportion of ZP-free eggs fertilized by mutant sperm were still at Met II in contrast to all eggs exhibiting 2nd polar body in controls, supporting that immediate post-fertilization timing defects could be contributing to embryo development failure. In this regard, incorporation of the time lapse technology in human IVF treatments has highlighted the relevance of the timing of post-fertilization events and early cleavage as robust predictors of both embryo development to blastocyst and implantation success (*Basile et al., 2015*; *Esbert et al., 2018*; *Meseguer et al., 2011*).

Our observations showing defects in meiotic resumption in eggs fertilized by mutant sperm opened the possibility that an impaired egg activation could be the cause of embryo development failure. In this regard, the well-established significance of egg $Ca^{2+}$ oscillations in predicting the developmental competence of mouse zygotes and their pivotal role in meiosis resumption (*Miyazaki, 2006*; *Wakai et al., 2019*), led us to analyze egg $Ca^{2+}$ oscillations following fertilization by mutant sperm. Results showing no difference in the pattern of oscillations nor in the time to the first peak which is indicative of gamete fusion confirmed immediate post-fertilization defects as responsible for egg development failure, dismissing alterations in $Ca^{2+}$ oscillations as the cause of embryo development impairment in the mutant mice.

Evidence showing that time is needed before the first embryonic cell division for activation of the egg DNA-repairing machinery (*Martin et al., 2019*; *Newman et al., 2022*) together with the known association between high levels of DNA fragmentation and abnormal embryonic development (*Marinaro, 2023*; *Nguyen et al., 2023*; *Simon et al., 2014*), raised the possibility that meiotic resumption failure may result from defects in sperm DNA integrity. In this regard, as spermatozoa harboring DNA damage have no mechanism for its repair but retain their ability to negotiate the female reproductive tract, reach the site of fertilization and fertilize oocytes, DNA-repairing activity relies on the oocyte once fertilization takes place (*González-Marín et al., 2012*; *Martin et al., 2019*), representing a critical step in the generation of viable embryos. Our findings showing significantly higher levels of DNA fragmentation in mutant than control cauda epididymal sperm support sperm DNA damage as the potential cause of the early post-fertilization defects. In agreement with this conclusion, several embryo kinetics studies showed a significant delay in the early stage of second polar body extrusion in human eggs fertilized by sperm with DNA fragmentation (*Casanovas et al., 2019*; *Esbert et al., 2018*; *Wdowiak et al., 2015*). Together, our observations underscore the intricate relationship between sperm DNA integrity and the regulatory role of CRISP proteins in shaping early embryonic outcomes.

How the lack of male CRISP1 and CRISP3 proteins could lead to higher levels of sperm DNA fragmentation? The finding that defects leading to embryo development in the mutant colony were present in epididymal sperm and that neither CRISP1 nor CRISP3 are expressed in the testes, supports the appearance of mutant sperm DNA fragmentation within the epididymis. In this regard, detection of these defects in cauda but not caput sperm supports that DNA integrity defects develop during epididymal transit and/or storage in agreement with the idea that the main pathways leading to DNA damage are triggered during epididymal transit (*Okada et al., 2020*; *Sakkas and Alvarez, 2010*; *Suganuma et al., 2005*), probably to allow elimination of defective sperm (*Gawecka et al., 2015*). Based on this, we can speculate that the lack of CRISP1 and CRISP3 in the epididymis renders sperm more susceptible to DNA degradation, i.e., by ROS. However, in spite of the commonly accepted association between DNA fragmentation and oxidative stress, our results did not show higher levels of ROS in mutant sperm, not favoring oxidative stress as the mechanism underlying DNA fragmentation in our mutant colony. These results differ from those reported for mice lacking epididymal antioxidant

GPX5 (Glutathione Peroxidase 5) which showed higher incidence of embryo development defects and increased sperm DNA fragmentation but accompanied by higher levels of ROS in mutant sperm (*Chabory et al., 2009*). Nevertheless, as these mice also exhibited changes in small RNAs within the epididymis (*Chu et al., 2020*), we do not exclude the possibility that the lack of CRISP1 and CRISP3 in the epididymis may also affect the profile of small RNAs known to change during epididymal transit and to be important for embryonic development (*Yuan et al., 2016*).

Previous reports showing induction of DNA fragmentation by exposure of mouse sperm to divalent cations ($Ca^{2+}$ and $Mn^{2+}$) in the presence of luminal fluids concluded that DNA fragmentation is largely controlled by the luminal fluids with a contribution from sperm that is acquired during maturation (*Gawecka et al., 2015*; *Shaman et al., 2006*). This, together with the fact that all CRISP are $Ca^{2+}$ channel regulators, opened the possibility that changes in $Ca^{2+}$ regulation within the epididymis could be responsible for the higher fragmentation levels observed in our mutant sperm. Consistent with this idea, our results showed a significant increase in DNA fragmentation when WT sperm were exposed to either mutant epididymal fluid or WT epididymal fluid in the presence of $Ca^{2+}$. These observations, together with the higher intracellular $Ca^{2+}$ levels detected in mutant sperm, support a dysregulation in $Ca^{2+}$ homeostasis in the epididymis and sperm as the main responsible for the observed sperm DNA integrity defects, probably through the activation of $Ca^{2+}$-dependent nucleases present within the epididymal fluid and/or sperm cells (*Shaman et al., 2006*; *Sotolongo et al., 2005*; *Boaz et al., 2008*; *Dominguez and Ward, 2009*).

In summary, our observations provide strong evidence supporting the contribution of the epididymis beyond the acquisition of sperm fertilizing ability, identifying CRISP1 and CRISP3 as novel male factors relevant for sperm DNA integrity and early embryo development. Moreover, to our knowledge, this is the first report showing the relevance of epididymal proteins for early post-fertilization events. Given the existence of homologous human CRISP proteins in the epididymis and sperm playing equivalent roles to their rodent counterparts (*Curci et al., 2020*; *Gonzalez et al., 2021*), it is possible that CRISP are also involved in human embryogenesis through similar mechanisms. Considering the incidence of sperm DNA fragmentation in male infertility (*Agarwal et al., 2020*; *Esteves et al., 2020*), we believe our findings not only provide key information on the impact of epididymal factors on mammalian embryo development but will also contribute to a better understanding, diagnosis, and treatment of human infertility.

## Materials and methods
### Animals and ethical approval
Adult males (3–5 months old) from C1/C3 DKO (*Curci et al., 2020*) or C1/C4 DKO colonies (*Carvajal et al., 2018*) and control young (3–5 weeks old) or adult (2–5 months old) females were used. Animals were maintained with food and water *ad libitum* in a temperature-controlled room with a 12:12 hr light:dark cycle. Approval for the study protocol was obtained from the CICUAL of Instituto de Biología y Medicina Experimental (IByME-CONICET) (protocol N° 26/2018). All protocols were conducted in accordance with the Guide for Care and Use of Laboratory Animals published by the National Institutes of Health (NIH).

**Key resources table**

| Reagent type (species) or resource | Designation | Source or reference | Identifiers | Additional information |
|---|---|---|---|---|
| Gene (*Mus musculus*) | *Crisp1* | Ensembl | ENSMUSG00000025431 | |
| Gene (*Mus musculus*) | *Crisp3* | Ensembl | ENSMUSG00000025433 | |
| Gene (*Mus musculus*) | *Crisp1* | Ensembl | ENSMUSG00000025431 | |
| Gene (*Mus musculus*) | *Crisp4* | Ensembl | ENSMUSG00000025774 | |
| Strain, strain background (*Mus musculus*) | C57BL/6J | IBYME Animal Facility, Buenos Aires, Argentina | | https://doi.org/10.1096/fj.202001406R |
| Strain, strain background (*Mus musculus*) | C57BL/6*DBA | IBYME Animal Facility, Buenos Aires, Argentina | | https://doi.org/10.1038/s41598-018-35719-3 |

*Continued on next page*

*Continued*

| Reagent type (species) or resource | Designation | Source or reference | Identifiers | Additional information |
|---|---|---|---|---|
| Chemical compound, drug | Equine chorionic gonadotropin | Zoetis, Buenos Aires, Argentina | Novormon | 5 UI |
| Chemical compound, drug | Human chorionic gonadotropin (hCG) | Zoetis, Buenos Aires, Argentina | Ovusyn | 5 UI |
| Chemical compound, drug | BSA | Sigma-Aldrich | A6003 | 3 mg/ml (capacitation medium) 1 mg/ml (KSOM médium) |
| Chemical compound, drug | Xylazine/ketamine | Holliday/Richmond Vet Pharma | | 10:100 mg/kg |
| Chemical compound, drug | Eosin B | Sigma-Chemical company | E-2629 | 0.5% (vol/vol) |
| Chemical compound, drug | Fluo-4 AM | Invitrogen | F14201 | 1 µM |
| Chemical compound, drug | Pluronic(R) F-127 Low UV | Invitrogen | P6867 | 0.02% (p/vol) |
| Chemical compound, drug | Vectashield | Vector | H-1000-10 | |
| Chemical compound, drug | CellROX-Green | Invitrogen | C10444 | 25 µM |
| Chemical compound, drug | Hoechst 33342 | Invitrogen | H1398 | 10 µg/ml |

### *In vivo* fertilization assays and *in vitro* embryo development

Males were caged individually for one night with superovulated females. For ovulation induction, females were treated with an i.p. injection of equine chorionic gonadotropin (5 UI, Zoetis, Buenos Aires, Argentina), followed by an i.p. injection of human chorionic gonadotropin (hCG; 5 IU, Zoetis, Buenos Aires, Argentina) 48 hr later. Mating was evaluated the following morning and considered successful by the presence of copulatory plugs. Eggs were then recovered from the oviducts, placed in KSOM médium (*Erbach et al., 1994*) supplemented with 0.1% (wt/vol) of bovine serum albumin (BSA), covered with paraffin oil (Ewe, Sanitas SA, Buenos Aires, Argentina), and incubated overnight at 37°C in an atmosphere of 5% (vol/vol) $CO_2$ in air. Eggs were considered fertilized when they reached the two-cell embryo stage. For evaluation of their development to blastocyst, two-cell embryos were incubated for an additional 3 days under the same conditions.

### Sperm transport and migration within the female tract

Male mice expressing a transgene for an acrosomal EGFP were mated with superovulated females to detect sperm within the oviduct, as previously described (*La Spina et al., 2016*; *Curci et al., 2020*). Briefly, a wild-type female subjected to superovulation was caged for 45 min with a transgenic male 12 hr after hCG administration. After 4 hr of detection of copulatory plug, the uterus and the oviducts were placed in KSOM medium supplemented with 0.3% (wt/vol) of BSA, mounted on slides, covered with coverslips and immediately observed under an Olympus IX83 microscope (Olympus Corp, Tokyo, Japan) at ×40. The number of fluorescent sperm within the oviduct was evaluated subjectively.

### Epididymal sperm collection and *in vitro* capacitation

Mouse sperm were recovered by incising the cauda epididymis in 150 µl of capacitation medium (*Fraser and Drury, 1975*; *Giaccagli et al., 2021*) supplemented with 0.3% (wt/vol) BSA, pH 7.3–7.5, allowing motile sperm to swim-out of the cauda for 10 min at 37°C in an atmosphere of 5% (vol/vol) $CO_2$ in air. For *in vitro* capacitation, aliquots of the swim-out suspension were added to 300 µl of capacitation medium to a final concentration of $1–10 \times 10^6$ spermatozoa/ml and incubated for 90 min under the same conditions.

### Intrauterine insemination

For intrauterine insemination young females were superovulated as previously described. Eight hours after hCG injection, the females were anesthetized with an i.p. injection of xylazine/ketamine (10:100 mg/kg), and an incision was made in the abdomen, exposing both uterine horns. Using a

syringe, 50 µl of either mutant or control swim-out sperm suspensions ($1 \times 10^7$ spermatozoa/ml) were introduced into one uterine horn followed by immediate ligation, whereas the remaining sperm suspension was introduced in the contralateral horn. After approximately 15 hr, oocytes were recovered from the ampulla, placed in KSOM medium, incubated at 37°C and 5% vol/vol $CO_2$, and the percentage of cells reaching the two-cell embryo stage was analyzed the following day. Two-cell embryos were then incubated for additional 3 days under the same conditions to evaluate their development to blastocyst.

## IVF assays

Gamete interaction assays were carried out as previously reported (*Curci et al., 2020*). Briefly, COC were collected from superovulated females 12–15 hr after hCG administration. When needed, cumulus cells were removed by incubating the COC in 0.3 mg/ml hyaluronidase (type IV) for 3–5 min and ZP was dissolved by treating the eggs with acid Tyrode solution (pH 2.5) for 10–20 s (*Nicolson et al., 1975*). COC were inseminated with a final concentration of $1–5 \times 10^5$ cells/ml and gametes were co-incubated for 3.5 hr at 37°C in an atmosphere of 5% (vol/vol) $CO_2$ in air. For gamete fusion assays, ZP-free eggs were inseminated with a final concentration of $1–5 \times 10^4$ cells/ml and gametes co-incubated for 1 hr under the same conditions. In all cases, eggs were recovered at the end of incubation, washed, fixed with 2% (wt/vol) paraformaldehyde in PBS and stained with 10 µg/ml Hoechst 33342 for evaluation of fertilization under epifluorescence microscope (×200). Eggs were considered fertilized when at least one decondensing sperm nucleus or two pronuclei were observed in the egg cytoplasm. Alternatively, ZP-intact or ZP-free eggs were recovered at the end of incubation and placed in KSOM medium for 24 hr to determine the percentage reaching the two-cell embryo stage or for 3 additional days for evaluation of eggs in blastocyst stage. To avoid sticking, ZP-free eggs were incubated in individual droplets. For evaluation of progression to different stages of embryo development, two-cell embryos obtained from COC were incubated for 3 days in KSOM and the percentage reaching each stage (i.e. 4/8 cells, morula or blastocyst) determined.

## Analysis of sperm functional parameters

Epididymal sperm concentration was determined using a hemocytometer. Viability was assessed by staining sperm with prewarmed 0.5% (vol/vol) eosin Y and dye exclusion (indicative of sperm viability) analyzed under light microscopy (×400). For progressive motility assessment, sperm suspensions (15 µl) were placed between prewarmed slides and coverslips (22 mm × 22 mm) to create a chamber with 30 mm depth and sperm movement was recorded by video microscopy under a light microscope (Nikon ECLIPSE E200; Basler acA-78075gc) at ×400 magnification for subsequent analysis. The percentage of progressive motile sperm was calculated by analyzing a minimum of 300 cells distributed in at least 20 different microscope fields.

## Oocyte Ca²⁺ oscillations

ZP-free eggs were incubated with 1 µM Fluo-4 AM, 0.02% (p/vol) pluronic acid and 15 µg/ml Hoechst 33342 in capacitation medium for 25 min at room temperature. Eggs were then extensively washed in fresh medium, mounted in 100 µl of medium covered with paraffin oil and analyzed on an Olympus IX83 Spinning Disk microscope (Olympus Corp, Tokyo, Japan) (×100), equipped with an environmental chamber sustaining a temperature of 37.5°C and 5% $CO_2$. Images were taken every 20 s. Basal $Ca^{2+}$ was recorded for 10 min. Then, *in vitro* capacitated sperm were added and image recording continued for at least 1.5 hr. In all cases, fertilization was analyzed by the presence of at least one decondensing sperm head within the ooplasm. Polyspermic eggs were excluded from the analysis. Intracellular $Ca^{2+}$ was determined in a single equatorial plane of each egg by measuring the fluorescence intensity using the ImageJ software (http://imagej.nih.gov/ij) and normalized to basal fluorescence.

## SCD assay

Sperm DNA integrity was assessed as described before (*Fernández et al., 2003*). Briefly, aliquots of 200 µl of sperm from a swim-out in capacitating medium were mixed with 1% low-melting-point aqueous agarose (to obtain a 0.7% final agarose concentration) at 37°C. In those cases in which caput sperm were used, cells were centrifuged for 3 min at 1500 rpm prior to the addition of agarose. Aliquots of 50 µl of the mixture were pipetted onto a coverslip and then placed over a glass slide

precoated with 0.65% standard agarose and left to solidify at 4°C for 10 min. Coverslips were carefully removed, and slides were immediately immersed horizontally in a tray with freshly prepared acid denaturation solution (0.08 N HCl) for 14 min at room temperature in the dark to generate restricted single-stranded DNA (ssDNA) motifs from DNA breaks. Then, proteins were removed by transfer of the slides to a tray with neutralizing and lysing solution 1 (0.4 M Tris, 0.8 M β-mercaptoetanol, 1% SDS, and 50 mM EDTA, pH 7.5) for 20 min at room temperature, which was followed by incubation in neutralizing and lysis solution 2 (0.4 M Tris, 2 M NaCl, and 1% SDS, pH 7.5) for 15 min at room temperature. Slides were thoroughly washed in TBE buffer (0.09 M Tris-borate and 0.002 M EDTA, pH 7.5) for 12 min, dehydrated in sequential 70%, 90%, and 100% ethanol baths (2 min each), and air dried. Cells were stained with Hoechst (10 µg/ml) in Vectashield (Vector Laboratories, Burlingame, CA) and halo surface analyzed by fluorescence microscopy on an Olympus IX83 Spinning Disk microscope (Olympus Corp, Tokyo, Japan) (×600, AN 1.42). Pictures of at least 200 sperm heads were taken and the halo area analyzed by ImageJ software (http://imagej.nih.gov/ij). Sperm DNA was considered fragmented when no halo was observed or when the halo area was smaller than twice the area corresponding to non-dispersed sperm.

## Analysis of ROS

ROS levels in sperm were measured by confocal microscopy. Briefly, after swim-out in capacitating medium without BSA, Hoechst 33342 (40 µg/ml) and CellROX-Green (25 µM) were added and sperm incubated under these conditions for 30 min at 37°C and 5% $CO_2$. Samples were then fixed with 4% paraformaldehyde for 10 min, exposed to 100 mM ammonium acetate (pH 9.0), centrifuged at 3000 rpm for 3 min two times and, finally, placed over a glass slide, mounted with glycerol and analyzed on an Olympus IX83 Spinning Disk microscope (Olympus Corp, Tokyo, Japan) (×600). Images of at least 200 sperm heads for each condition were analyzed and fluorescence intensity calculated by ImageJ software (http://imagej.nih.gov/ij).

## Incubation of sperm with epididymal fluids

After 10 min of swim-out in capacitating medium without BSA, sperm were centrifuged 2 min at 1500 rpm and the recovered supernatants containing diluted cauda fluid stored at 37°C in an atmosphere of 5% (vol/vol) $CO_2$ in air. Sperm were then washed two times with 1 ml of capacitating medium without BSA followed by a 2-min centrifugation at 1500 rpm. After the second centrifugation, cauda fluids from either the same or the other genotype were added and incubation continued for 1 hr at 37°C in an atmosphere of 5% (vol/vol) $CO_2$. For evaluation of the effect of $Ca^{2+}$, sperm were incubated as described above in the presence of 10 mM $Ca^{2+}$ prepared from a 100× $CaCl_2$ stock solution.

## Sperm intracellular $Ca^{2+}$ measurement

Cytoplasmic $Ca^{2+}$ levels in sperm were measured by flow cytometry as previously described (*Brukman et al., 2016*; *Curci et al., 2020*). Briefly, after 60 min of incubation in capacitation medium, sperm were loaded with 2 mM of Fluo-4 AM (Invitrogen, Carlsbad, California, USA) diluted in 10% (wt/vol) of Pluronic F-127 (Invitrogen) and incubated for an additional 30 min. Samples were washed to remove the excess of probe, resuspended in BSA-free medium, and exposed to 2.5 µg/ml of propidium iodide (PI) just before measurement. Fluorescence was detected using a BD FACSCantoTM II analyzer following the manufacturer's indications and at least 10,000 events were analyzed per sample. Data analysis was performed by FlowJo 10 software (FlowJo LLC, Ashland, OR, USA). In each condition, the fluorescence mean was normalized to basal Fluo-4-AM fluorescence.

## Statistical analysis

Data represent the mean ± SEM of at least three independent experiments and '*n*' indicates the number of animals analyzed in each group for all experiments except for oocyte $Ca^{2+}$ oscillations where '*n*' indicates the number of oocytes analyzed. Calculations were performed using the Prism 8.0 software (GraphPad Software, La Jolla, CA). Comparisons between two experimental groups were analyzed by one-way Student *t*-test whereas comparisons among three or more groups were analyzed by two-way ANOVA followed by Fisher LSD for embryo development progression and Holm–Sidak's for both resumption of meiosis and sperm intracellular $Ca^{2+}$. In those cases in which data did not meet the assumptions required to perform two-way ANOVA (i.e. caput sperm DNA fragmentation, ROS

levels, and incubation with fluids), non-parametric Kruskal–Wallis followed by Dunn's test were used. In all cases, differences were considered significant at a level of $p < 0.05$.

## Acknowledgements

The authors would like to thank Dr Cuasnicu laboratory members, especially Dr Debora Cohen, for their helpful comments and critical input as well as Fundación Williams and Fundación Barón for supporting the incorporation and maintenance of institutional equipment.

## Additional information

### Funding

| Funder | Grant reference number | Author |
|---|---|---|
| Consejo Nacional de Investigaciones Científicas y Técnicas | PIP 2022 | Mariana Weigel Muñoz Patricia S Cuasnicu |
| Agencia Nacional de Promoción de la Investigación, el Desarrollo Tecnológico y la Innovación | PICT 2019, No 3588 | Mariana Weigel Muñoz |
| Agencia Nacional de Promoción de la Investigación, el Desarrollo Tecnológico y la Innovación | PICT 2021, No 00765 | Patricia S Cuasnicu |

The funders had no role in study design, data collection, and interpretation, or the decision to submit the work for publication.

### Author contributions

Valeria Sulzyk, Conceptualization, Data curation, Formal analysis, Investigation, Methodology, Project administration, Software, Supervision, Validation, Visualization, Writing – original draft, Writing – review and editing; Ludmila Curci, Data curation, Formal analysis, Investigation, Methodology, Validation; Lucas N González, Methodology, Software, Visualization, Writing – review and editing; Abril Rebagliati Cid, Methodology, Writing – review and editing; Mariana Weigel Muñoz, Conceptualization, Data curation, Formal analysis, Funding acquisition, Investigation, Methodology, Project administration, Resources, Supervision, Validation, Visualization, Writing – original draft, Writing – review and editing; Patricia S Cuasnicu, Conceptualization, Data curation, Formal analysis, Funding acquisition, Investigation, Methodology, Project administration, Resources, Visualization, Writing – original draft, Writing – review and editing

### Author ORCIDs

Valeria Sulzyk ⬮ https://orcid.org/0009-0008-0170-8023
Mariana Weigel Muñoz ⬮ https://orcid.org/0000-0002-3320-0338
Patricia S Cuasnicu ⬮ https://orcid.org/0000-0002-4784-488X

### Ethics

Approval for the study protocol was obtained from the CICUAL of the Instituto de Biología y Medicina Experimental (IByME-CONICET) (protocol N° 26/2018). All protocols were conducted in accordance with the Guide for Care and Use of Laboratory Animals published by the National Institutes of Health (NIH).

Reviewer #1 (Public review): https://doi.org/10.7554/eLife.97105.3.sa1
Reviewer #2 (Public review): https://doi.org/10.7554/eLife.97105.3.sa2
Author response https://doi.org/10.7554/eLife.97105.3.sa3

## Additional files

### Supplementary files
Supplementary file 1. Analysis of different parameters in fresh and capacitated cauda epididymal sperm.

MDAR checklist

### Data availability
All data generated or analysed during this study are included in the manuscript and supporting files; source data files have been provided for all figures.

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
