## [Editor Report · eLife Assessment]

This **valuable** study reports that epididymal proteins are required for embryogenesis after fertilization. The data presented are generally supportive of the conclusion and considered **solid**. This work will be of interest to reproductive biologists and andrologists.

---

## [Referee Report · Reviewer #1 (Public review)]

Summary:

The main observation that the sperm from CRISP proteins 1 and 3 KO lines are post-fertilization less developmentally competent is convincing. The data showing progressive acquisition of the sperm defects during epididymal transport and the exchange fluid studies showing the altered epididymal environment are important. However, the molecular characterization of the mechanism(s) that leads to these defects requires additional studies.

Strengths:

The generation of these double mutant mice is valuable for the field. Moreover, the fact that the double mutant line of Crisp 1 and 3 is phenotypically different from the Crisp 1 and 4 line suggests different functions of these epididymis proteins. The methods used to demonstrate that developmental defects are largely due to post-fertilization defects are also a considerable strength. The initial characterization that these sperm have altered intracellular Ca2+ levels, and increased rates of DNA fragmentation are valuable. The increase fragmentation of control sperm DNA when exposed to mutant epididymal fluid is significant and an excellent platform for future studies.

Weaknesses:

The study is mechanistically incomplete because evidence of how these proteins alter the environment is not shown. What are the target(s) of these proteins that result in increased Ca2+?

---

## [Referee Report · Reviewer #2 (Public review)]

Summary:

The study highlights the role of CRISP1 and CRISP3, two epididymal proteins, in early embryo development through DNA integrity. The authors demonstrate that C1/C3 DKO sperm exhibit defects in the DNA integrity, probably due to Ca2+ dysregulation in the epididymis. However, direct evidence for this mechanism requires further experiments. The finding of the involvement of the epididymal environment in embryogenesis is significant, but some results on sperm fertilizing ability of C1/C3 DKO mice were similar to the previous report. Thus, this point raises concern about the perspective of novelty.

Strengths:

The authors demonstrate that CRISP1 and CRISP3 regulate Ca2+ in the epididymal fluid, and loss of CRISP1 and CRISP3 disrupts Ca2+ regulation in the epididymal fluid, leading to sperm DNA fragmentation and impaired embryonic development after fertilization. This proposed mechanism is both novel and intriguing, offering valuable insights into the epididymal control of sperm quality.

Weaknesses:

The evidence supporting the mechanism of CRISP1 and CRISP3 in calcium regulation within epididymis and its contribution to the sperm DNA damage remains limited.

Major comments:

The data provided in this manuscript (Figure 2A and B) appear to overlap with data in previously published paper (PMID:33037689), despite differences in the duration of *in vivo* fertilization after mating. The results in both studies show similar findings, raising concerns about potential data redundancy.

As shown in Figure 6A, while wild-type sperm were exposed to the epididymal fluid of C1/C3 DKO mice, the wild-type sperm exhibited DNA fragmentation. Additionally, when wild-type sperm were exposed to the epididymal fluid of wild-type mice with 10 mM Ca2+, DNA fragmentation is still observed. Therefore, the authors conclude that the DNA fragmentation in C1/C3 DKO sperm is due to the increased level of the Ca2+. However, the connection between the DNA damage in wild-type sperm exposed to the epididymal fluid of C1/C3 DKO mice and the increased levels of Ca2+ remains unclear. To clarify this, it is suggested that intracellular calcium levels in the wild type sperm should be analyzed before and after exposure to the epididymal fluid of C1/C3 DKO mice (or before and after adding 10 mM Ca2+ into wild-type fluid). Furthermore, the author should explain detailed information on epididymal fluid collection, because Ca2+ levels vary between different sections of the epididymis.

In lines 321-323, the authors mention the selection system of the female reproductive tract that only allows high-quality sperm to reach the eggs (Cummins and Yanagimachi 1982), but this paper is not listed in the bibliography. It is important to ensure proper referencing.

The discussion section is too long and difficult to follow well because there is redundancy of the results in many parts. It is recommended to shorten it by focusing only on relevant and important information.

---

## [Author Response]

The following is the authors’ response to the previous reviews.

**Public Reviews:**

**Reviewer #1 (Public Review):**
Summary:The main observation that the sperm from CRISP proteins 1 and 3 KO lines are postfertilization less developmentally competent is convincing. However, the molecular characterization of the mechanism that leads to these defects and the temporal appearance of the defects requires additional studies.

We thank the reviewer for the valuable comments. As requested, additional experiments were carried out to analyze both the molecular mechanisms and the temporal appearance of the observed defects. Our results showed that DNA integrity defects appear during epididymal maturation and/or storage (see Figure 5B), that the epididymal fluid contributes to sperm DNA fragmentation defects (See Figure 6A) and that these defects seem not to be due to an increase in oxidative stress (Figure 5C) but rather to a dysregulation in Ca^2+^ homeostasis within the epididymis (Figure 6A,B).

Strengths:The generation of these double mutant mice is valuable for the field. Moreover, the fact that the double mutant line of Crisp 1 and 3 is phenotypically different from the Crisp 1 and 4 line suggests different functions of these epididymis proteins. The methods used to demonstrate that developmental defects are largely due to post-fertilization defects are also a considerable strength. The initial characterization of these sperm has altered intracellular Ca^2+^ levels, and increased rates of DNA fragmentation are valuable.

We thank the reviewer for the positive comments on our work.

Weaknesses:The study is mechanistically incomplete because there is no direct demonstration that the absence of these proteins alters the epididymal environment and fluid, wherein during the passage through the epididymis the sperm become affected. Also, a direct demonstration of how the proteins in question cause or lead to DNA damage and increased Ca^2+^ requires further characterization.

The new experiments included in the revised version (see Figure 6A) showed that exposure of control WT sperm to epididymal fluid form mutant mice leads to an increase in sperm DNA fragmentation levels, confirming that the absence of CRISP1 and CRISP3 alters the epididymal fluid wherein the sperm become affected. In addition, new observations showing that WT sperm exposed to WT epididymal fluid in the presence of Ca^2+^ also exhibit higher DNA fragmentation levels (Figure 6A) together with the finding that mutant sperm exhibit higher intracellular Ca^2+^ levels (Figure 6B) but no higher levels of ROS, strongly support a dysregulation in Ca^2+^ homeostasis within the epididymis and sperm as the main responsible for DNA integrity defects.

**Reviewer #2 (Public Review):**
The authors showed that CRISP1 and CRISP3, secreted proteins in the epididymis, are required for early embryogenesis after fertilization through DNA integrity in cauda epididymal sperm. This paper is the first report showing that the epididymal proteins are required for embryogenesis after fertilization. However, some data in this paper (Table 1 and Figure 2A) are overlapped in a published paper (Curci et al., FASEB J, 34,15718-15733, 2020; PMID: 33037689). Furthermore, the authors did not address why the disruption of CRISP1/3 leads to these phenomena (the increased level of the intracellular Ca^2+^ level and impaired DNA integrity in sperm) with direct evidence. Therefore, if the authors can address the following comments to improve the paper's novelty and clarification, this paper may be worthwhile to readers.

We thank the reviewer for the constructive comments. Regarding the data included in Table 1 and Figure 2A, it is important to note that Table 1 includes data on embryo development corresponding to C1/C4 DKO mice not published before in which the data on embryo development corresponding to C1/C3 DKO was used as simultaneous control. Figure 2A showed *in vivo* fertilization results at short times after mating (4h instead of 18 h) that have been neither reported before.

Regarding studies to address why the disruption of CRISP1 and CRISP3 leads to defects in DNA integrity and Ca^2+^ levels, we have carried out new experiments showing that mutant sperm do not exhibit higher levels of ROS (see Figure 5C), not favoring oxidative stress as the mechanism underlying mutant sperm defects. In addition, we found that DNA integrity defects develop during epididymal transit (Figure 5B) and that exposure of WT sperm to epididymal fluid from mutant mice leads to an increase in sperm DNA fragmentation levels (Figure 6A), confirming that the absence of CRISP1 and CRISP3 alters the epididymal fluid. Finally, our new results showing that WT sperm exposed to WT epididymal fluid in the presence of Ca^2+^ also exhibit higher DNA fragmentation levels (Figure 6A) together with the higher intracellular Ca^2+^ levels detected in mutant sperm (Figure 6B) strongly support a dysregulation in Ca^2+^ homeostasis within the epididymis and sperm as the main responsible for DNA integrity defects.

**Recommendations for the authors:**

**Reviewer #1 (Recommendations For The Authors):**
Overall comments:This manuscript investigates the mechanisms whereby the absence of the epididymal CRISP proteins 1 and 3 (Cysteine-Rich Secretory Proteins) causes infertility and lower embryo developmental rates. This strain's infertility seems to have a post-fertilization origin because the rates of *in vivo* fertilization are like the controls, but the development to the blastocyst stage is decreased. The results of this study show that (1) mutant sperm viability, progressive motility, and morphology are normal;(2) *in vivo* fertilization rates are comparable to controls, but embryo development is reduced;(3) *in vitro* fertilization studies found reduced fertilization rates and activation rates even in zona-free studies;(4) additional functional studies showed increased rates of DNA fragmentation and elevated Ca^2+^ levels in mutant sperm.The results presented are credible and hint that the epididymis might play a role before and after fertilization and directly affect embryo development. However, the study is mechanistically incomplete, as there is no direct demonstration that the absence of these proteins alters the epididymal environment and fluid, wherein the passage through the epididymis the sperm become functionally defective, and whether mutant or control epididymal fluid or purified CRISP proteins can change, either reduce or overcome, respectively, the developmental competence of the control or mutant sperm and induce functional changes in the counterpart sperm. In summary, the main observation that the sperm from CRISP proteins 1 and 3 KO lines are post-fertilization less developmentally competent is significant and important, but the molecular characterization of the defects and the temporal appearance of defects requires additional studies.Specific comments:(1) Introduction.It is too long. The description of the function of the epididymis should be reduced. The functional properties of the Crisp genes should also be substantially shortened.

As requested, the Introduction has been revised and descriptions of the epididymis and CRISP have been shortened

(2) Results.• Lines 140 to 142. Remove these initial lines. Start directly addressing the results of the C1/C3 strain, which is the mutant under consideration here. Referring to the C1/C4 results detracts from the focus of the study.

As suggested by the reviewer, lines 140 to 142 have been removed.

• Table 1. Move the two-cell embryo line to the top of the Table and place the Blastocyst line below it. This organization is the conventional method to present this type of data.

As suggested, the order of the lines in Table 1 has been modified to align with the conventional presentation method.

• Figures 1 and 2A and B data are solid and support the notion that enough sperm reach the site of fertilization, and that the sperm are defective in their capacity to support embryo development. Figures 2C and D have interesting data, although additional information would strengthen these results. The authors concluded that the sperm were defective in the epididymis. Where in the epididymis? These sperm were all from the cauda. Could the authors collect sperm from the upper portion of the cauda, or midportion, and compare if the defects manifest gradually?

We appreciate this interesting and appropriate comment from the reviewer. In this regard, all the studies in our work were carried out using sperm from the whole cauda epididymis, the reason why we could not answer where defective sperm appear in the epididymis. In view of this, we have now conducted a comparative DNA fragmentation analysis between caput and cauda sperm from both genotypes. Our findings indicate that while cauda mutant sperm showed once again higher DNA fragmentation levels than controls, caput sperm exhibited levels of DNA damage not significantly different between genotypes. These results confirm that defects in DNA appear following sperm passage through the epididymal caput, supporting the hypothesis that defects in DNA fragmentation manifest during sperm transit through the epididymis and /or during storage in the cauda. These results have been included in the revised version of the manuscript (see lines 235-240/Figure 5B of the revised version)

• Figure 3 displays the results of *in vitro* fertilization, either COCs A-C or zona-free fertilization D-F. The results are important and differ from those produced by fertilization *in vivo*. The authors indicate that these confirm that the *in vivo* conditions overcome *in vitro* defects. However, this study never addresses the reason behind it. Is there less expression of proteins related to these functions, or the function of some proteins is compromised? The authors should advance a hypothesis or a rationale to explain these results.

As indicated by the reviewer, our results showed differences between the fertilization rates observed for mutant mice under *in vivo* and *in vitro* conditions, as previously observed for all our single and multiple KO models (Da Ros et al., 2008; PMID: 18571638, Brukman et al., 2016; PMID: 26786179, Weigel Muñoz, 2018; PMID: 29481619, Ernesto et al., 2015; PMID: 26416967, Carvajal et al,. 2018; PMID: 30510210) and also reported by other groups (Okabe et al., 2007; PMID: 17558467). In this regard, it has been well established that, although millions of sperm are ejaculated into the female tract, only a few (approximately one per oocyte) reach the fertilization site (i.e. the ampulla) (Cummins and Yanagimachi, 1982; doi:10.1002/mrd.1120050304). This efficient selection system by the female reproductive tract leads to the arrival of only the best sperm at the fertilization site, even in males with reproductive deficiencies, thereby “masking” sperm defects that can be detected under *in vitro* conditions due to the competition between good and bad quality sperm for the egg. Thus, although we can not exclude other mechanisms to explain the commonly observed differences between *in vivo* and *in vitro* fertilization rates, our rationale is that the natural and efficient sperm selection process that takes place within the female reproductive tract masks sperm defects that can, otherwise, be detected under the competitive *in vitro* conditions. This explanation is now included in the discussion of the revised version of the manuscript (see lines 320-325).

• Data in Figures 4 and 5 support the interpretation of the authors. However, it is necessary to establish the level of oxidative stress in the mutant sperm vs. the controls. Also, a question to explore is for how long does the sperm need to reside in that mutant environment to start undergoing the DNA fragmentation reported?

In response to the valuable request from the reviewer regarding the level of oxidative stress in sperm, we have analyzed reactive oxygen species (ROS) levels in mutant and control epididymal sperm. Our results showed that ROS levels in mutant sperm were not higher than those observed in the control group, supporting the idea that mechanisms other than oxidative stress may be leading to the increased DNA fragmentation observed in mutant sperm. These results are now included in the revised version of the manuscript (see Figure 5C).

Regarding the question on how long the sperm need to reside in the mutant environment to undergo DNA fragmentation, recent experiments carried out in response to this reviewer in which we analyzed DNA fragmentation in caput sperm led us to conclude that DNA fragmentation develops during epididymal transit and/or storage in the cauda. While these observations do not precisely define the time within the epididymis that sperm require for exhibiting DNA fragmentation, our additional new *in vitro* experiments analyzing the effect of epididymal fluids on sperm DNA integrity showed that exposure of WT sperm to DKO fluid for only 1 hr already leads to an increase in DNA fragmentation (see Figure 6A of the revised manuscript), suggesting that sperm do not need long periods within the mutant environment to be affected.

(3) The length of the Discussion section should be shortened, especially by not recapitulating data presented in the Results section.

As requested by the reviewer, sections recapitulating results have been modified.

Minor comments:(1) The sentence in lines 171 and 172 is unclear, "However, despite the short time after mating, once again, the *in vivo* fertilized eggs corresponding to the mutant group exhibited clear defects to reach the blastocyst stage *in vitro* compared to controls." What do the authors mean by short time? It is the expected time, correct?

It is well established that after copulatory plug formation, most oocytes are fertilized within 2 to 8 hours, with fertilization rates that increase over time: 0–5% at 1.5 hours post-mating; 40% at 4 hours post-mating and more than 90% at 7 hs after mating (Muro et al., 2016; PMID: 26962112, La Spina et al., 2016; PMID: 26872876). In order to examine whether the embryo development defects observed for mutant mice were due to a delayed arrival of sperm to the ampulla, we decided to analyze the percentage of fertilized eggs recovered from the ampulla at “short times” (4 hs) after mating to avoid the possibility that the prolonged stay of sperm within the female tract corresponding to the usual “overnight mating” schedule could be giving defective sperm enough time to reach the ampulla and, finally, fertilize the eggs (i.e. delayed fertilization). Our results showed that, despite the expected lower fertilization rates observed for both control and mutant males when analyzed just 4 hs after mating, the fertilized eggs corresponding to the mutant group were still exhibiting clear defects to develop into blastocysts compared to controls, not favoring the idea that embryo development defects were due to a delayed fertilization. The sentence in lines “171 and 172” has been modified in the revised version of the manuscript to better explain this conclusion (see lines 152-155 of the revised version).

(2) Line 177. Mutant epididymal sperm already carry defects leading to embryo development failure. Under this subheading, the authors compare within the same female the ability of mutant and control sperm delivered into different horns to support fertilization and embryo development. They show that the embryo development induced by mutant sperm is diminished vs. controls under very similar conditions, confirming the previous results of post-fertilization failure. The data also answers the question raised by the authors of whether the fertilization defects appear during or after epididymal transit; the interpretation of the results is the functional defects in the sperm are present before the transport into the female tract. Important unaddressed questions are, could these defects begin even earlier before arriving at the cauda? Did the authors try to incubate the mutant sperm with the epididymal fluid of WT mice to examine if the sperm defects could be rescued? The opposite experiment could also be performed, where WT sperm are incubated with the epididymal fluid of mutant mice, and the treated sperm examined for altered Ca^2+^ levels or DNA fragmentation.

First of all, we would like to clarify that our question about whether the fertilization defects appear “during or after epididymal transit” was in fact referring to whether defects appear during epididymal maturation or later on, at the moment of ejaculation. In this regard, our *in vivo* and *in vitro* fertilization studies allowed us to conclude that defects were already present in epididymal sperm without excluding the possibility that additional defects could appear at the vas deferens or at the moment of ejaculation due to the contribution of seminal plasma secretions.

Regarding whether sperm defects could appear even earlier before arriving to the cauda, we have now analyzed DNA fragmentation defects in caput vs cauda both mutant and control sperm observing differences between genotypes only for cauda sperm. Based on these observations, we conclude that DNA integrity defects appear within the epididymis after sperm passage through the caput either when sperm reach the corpus or the cauda epididymis, or during their storage within the cauda region.

Also, as suggested by the reviewer, we incubated *in vitro* WT sperm with epididymal fluid from DKO mice (and vice versa) and then analyzed DNA fragmentation levels. Results showed that exposure of control sperm to the mutant epididymal fluid for 1 hr significantly increased DNA fragmentation levels. When mutant sperm (exhibiting higher levels of DNA fragmentation than control sperm), were exposed to epididymal fluid from WT mice, no differences between groups were observed. Together, these results confirm both that the epididymal fluid from mutant mice contributes to the higher DNA fragmentation levels detected in mutant sperm, and that normal epididymal fluid would not be able to rescue the DNA fragmentation present in mutant cells. These results are now included in the revised version of the manuscript (see Figure 6A).

(3) Lines 203 to 216. In these paragraphs the authors indicate "that mutant sperm had a lower percentage of fertilization and lower rates of blastocysts (Figure 3D, E), indicating that defects in egg coat penetration were not responsible for embryo development failure. Later, they indicated that a few eggs fertilized by mutant sperm failed to activate. It is shown that Ca^2+^ oscillations are normal, indicating that the defects lie elsewhere. Could the authors propose a mechanism based on their sperm DNA defects?

As described in the Result and Discussion sections of the original manuscript, we decided to investigate the existence of possible defects in sperm DNA fragmentation based on evidence indicating that delays in early embryo development may result from the time taken by the egg to repair damaged paternal DNA (Esbert *et al*., 2018; PMID: 30259705, Newman *et al.*, 2022; PMID: 34954800, Nguyen *et al*., 2023; PMID: 37658763). In this regard, it is known that time is needed before the first embryonic cell division for activation of the egg DNA repairing machinery (Martin et al., 2019; PMID: 30541031, Newman et al., 2022; PMID: 34954800) and that increased sperm DNA damage may necessitate more time for repair by the oocyte (Martin et al., 2019; PMID: 30541031, Newman et al., 2022; PMID: 34954800). Based on this, we decided to examine possible DNA damage in sperm. Our finding that, in fact, sperm DNA fragmentation was clearly increased in mutant sperm led us to propose that delays in early embryo development in our mutant colonies may result from the time required by the egg to repair sperm DNA fragmentation.

(4) The demonstration that C1/C3 sperm have abnormal rates of DNA fragmentation and Ca^2+^ levels is significant. Additional studies would strengthen the findings reported here. For example, what are the levels of oxidative stress in these sperm? Are there other changes related to oxidative stress? Performing a TUNNEL assay will strengthen the notion of DNA damage demonstrated here with the chromatin dispersion assay.

As mentioned previously, we analyzed oxidative stress by evaluating ROS levels in control and mutant sperm observing no differences between genotypes. These results have been included in the revised version of the manuscript (See Figure 5C). We appreciate the suggestion of performing TUNNEL assay for future studies.

**Reviewer #2 (Recommendations For The Authors):**
Major comments:(1) There are some reports small RNAs gained during the epididymal transition of sperm are essential for embryonic development (e.g., Conine et al., Dev Cell, 46, 470480, 2018; PMID: 30057276), suggesting that the luminal changes in Crisp1/3 double KO (dKO) epididymis lead to the phenotype in this study. In fact, there is no evidence whether CRISP1/CRISP3 secreted from an epididymis exists in cauda epididymal sperm and directly controls the observed phenomena. Also, the authors wrote there is no strong evidence to exclude the possible role of small RNA in Crisp1/3 dKO sperm (lines 370-372). Therefore, it is at least necessary to measure small RNA abundance in dKO mice.

As mentioned by the reviewer and as cited in our manuscript, there is a report indicating that the small RNAs gained during epididymal transit may play a role in embryonic development (Conine et al., 2018; PMID: 30057276). However, the need of small RNAs for embryonic development still remains a topic of debate (Wang et al. 2020; PMCID: PMC7799177). In this regard, clear evidence indicating that sperm DNA fragmentation is associated with embryo development defects together with the increase in sperm DNA fragmentation levels observed in mutant sperm support sperm DNA damage as one of the causes leading to the observed phenotype in our mutant mice. Moreover, recent experiments carried out in response to Reviewer 1 comments revealed that exposure of control sperm to epididymal fluid from mutant mice significantly increases DNA fragmentation levels, confirming that the absence of CRISP1 and CRISP3 proteins in epididymal fluid contributes to sperm DNA damage in mutant sperm. Finally, whereas oxidative stress might also lead to embryo development impairment as mentioned in our original manuscript, recent evaluation of ROS levels in control and mutant sperm carried out in response to Reviewer 1’s comments did not show higher ROS levels in mutant sperm. Thus, although as mentioned in the manuscript, we do not exclude the possibility that small RNAs may also contribute to embryo development defects, our observations support DNA fragmentation and a dysregulation in Ca^2+^ homeostasis within the epididymis and sperm as the main responsible for embryo development failure in our mutant males. The experiments using epididymal fluid (Figure 6A) and those evaluating ROS levels (Figure 5C) have been included in the revised version of the manuscript and discussed accordingly.

(2) Lines 245-248 and 354-374: According to Figure 5C, the intracellular Ca^2+^ level significantly increased in Crisp1/3 dKO sperm compared to control. The author hypothesized that this increase could destroy sperm DNA integrity, causing defects in early embryogenesis. However, the authors did not show the direct evidence.Specifically, as CRISP1 inhibits CatSper (line 95), the authors believed the increased Ca^2+^ level in Crisp1/3 dKO sperm was observed. Crisp1/3 dKO and Crisp1/4 dKO mice share the disruption of Crisp1, but the phenotype is totally different. Thus, the authors should also examine the CatSper activity in Crisp1/3 dKO sperm.

We appreciate the reviewer's insightful comments. In this regard, whereas C1/C3 and C1/C4 DKO colonies shares the disruption of Crisp1, the intracellular Ca^2+^ levels in these two colonies are different as no increase in sperm intracellular Ca^2+^ was detected in Crisp C1/C4 DKO mice. Thus, this difference in intracellular Ca^2+^ levels might explain the different embryo development phenotype observed in our two DKO colonies. In this regard, our results revealed that sperm intracellular Ca^2+^ levels are different depending on the Crisp gene being deleted. Whereas the lack of Crisp1 did not affect intracellular sperm Ca^2+^ levels (Weigel Munoz et al, 2018; PMID: 29481619), there was an increase in Ca^2+^ levels in CRISP2 KO sperm (Brukman et al., 2016; PMID: 26786179) and a decrease in sperm when Crisp4 was deleted (Carvajal 2019, Ph.D Thesis). Thus, although the ability of CRISP3 to regulate sperm Ca^2+^ channels has not yet been reported, the existence of functional compensations between homologous CRISP members (Curci et al., 2020; PMID: 33037689) makes it complicated to draw straightforward conclusions based on the behavior of each individual protein in Ca^2+^ regulation. In fact, while the lack of CRISP1 and CRISP4 does not affect sperm Ca^2+^ concentration (Carvajal 2019, Ph.D Thesis), the simultaneous lack of CRISP1 and CRISP3 produced an increase in Ca^2+^ levels and the lack of the four CRISP proteins showed a decrease in the intracellular levels of the cation after capacitation (Curci et al, 2020). Based on these observations, we conclude that the absence of CRISP1 may or may not lead to altered intracellular Ca^2+^ levels depending on the other simultaneously-deleted gene/s.

The authors make a hypothesis that the increased Ca^2+^ level may lead to damaged DNA integrity by citing a published paper (lines 360-363). In the published paper, the authors examined the influence of the luminal fluid of the epididymis and vas deference on sperm chromatin fragmentation (Gawecka et al., 2015). However, they did not mention the increased DNA fragmentation in epididymal sperm when these sperm were incubated with Ca^2+^ or Mn2+. So, the authors' hypothesis is over discussion. Thus, the correlation between the intracellular Ca^2+^ level and DNA integrity in sperm is still unclear. So, the authors should show why the increased Ca^2+^ level leads to DNA fragmentation with direct evidence.

We appreciate the reviewer’s comment regarding the work by Gawecka et al., (2015), and the opportunity to clarify the proposed mechanism underlying our observations. In the above mentioned paper, the authors reported that when mouse epididymal or vas deferens sperm were incubated with divalent cations (Ca^2+^ and Mn^2+^) in the presence of luminal fluid, they were induced to degrade their DNA in a process termed sperm chromatin fragmentation (SCF). The fact that both the ejaculated and epididymal mutant sperm used in our studies had been exposed to epididymal fluid lacking CRISP proteins known to regulate sperm Ca^2+^ channels, opened the possibility that changes in Ca^2+^ levels within the epididymal fluid and/or sperm could be responsible for the higher DNA fragmentation levels observed in mutant cells. In this regard, it is important to note that, as requested by Reviewer 1, we performed additional *in vitro* experiments in which WT epididymal sperm were exposed to mutant or WT epididymal fluid in the presence or absence of Ca^2+^ and DNA fragmentation analyzed at the end of incubation. Results showed a significant increase in DNA fragmentation in WT sperm exposed to either mutant epididymal fluid or WT fluid in the presence of Ca^2+^ (Figure 6A). We believe these observations together with the higher intracellular Ca^2+^ levels detected in DKO sperm (Figure 6B) provides strong evidence supporting changes in Ca^2+^ homeostasis in the epididymis and sperm as the main responsible for the observed sperm DNA integrity defects. This could be mediated by the activation of Ca^2+^-dependent nucleases present within the epididymal fluid and/or sperm cells as previously suggested (Shaman *et al.*, 2006; PMID: 16914690, Sotolongo *et al.*, 2005; PMID: 15713834, Boaz *et al.*, 2008; PMID: 17879959, Dominguez and Ward, 2009; PMID: 19938954). These observations have now been included and discussed in the revised version of the manuscript (see lines 245-265 and 427-439).

Minor Comments:(3) Standards for measuring rates should be clarified, such as two-cell rates are determined by dividing the number of two-cell embryos by the total number of eggs.

As requested, standards for measuring rates have now been clarified in the corresponding figure legends